# PAC-Bayes Bounds for Multivariate Linear Regression and Linear Autoencoders

## Abstract

Linear Autoencoders (LAEs) have shown strong performance in state-of-the-art recommender systems. However, these impressive results are mainly based on experiments, with little theoretical support. This paper investigates the generalizability – a theoretical measure of model performance in statistical machine learning – of multivariate linear regression and LAEs. We first propose a PAC-Bayes bound for multivariate linear regression, which is generalized from an earlier PAC-Bayes bound for multiple linear regression by (Shalaeva et al., 2020), and outline sufficient conditions that ensure its theoretical convergence. We then apply this bound to LAEs by showing that LAEs can be viewed as constrained multivariate linear regression on bounded data, and develop practical methods for minimizing the bound, addressing the calculation challenges posed by the constraints. Experimental results demonstrates the non-vacuousness of our bound for LAEs.

## 1. Introduction

In recent years, simple (linear) recommendation models have consistently demonstrated impressive performance, often rivaling deep learning models (Dacrema et al., 2019; Jin et al., 2021; Mao et al., 2021), especially for the implicit setting, where interactions are inferred from user behavior (e.g., clicks or purchases). In particular, linear autoencoders (LAEs) such as EASE (Steck, 2019) and EDLAE (Steck, 2020) have shown a surprising edge over widely used matrix factorization (MF) methods such as ALS (Hu et al., 2008).

Despite their power and widespread use, linear autoencoders, particularly in the context of recommendation systems, remain theoretically underexplored. Recommendation research has understandably focused on performance evaluation to compare models, but issues such as weak baselines

and unreliable sampled metrics often make these evaluations difficult to reproduce (Dacrema et al., 2019; Cremonesi & Jannach, 2021). A recent study attempted to provide a theoretical comparison between linear recommendation models, such as matrix factorization and LAE, using spectral analysis, showing that both approaches "reduce" the singular values of the original user-item data matrix $R$, albeit in different ways (Jin et al., 2021). Another related study investigates the loss landscape of low-rank LAEs, characterizing their critical points through the smooth submanifold theory (Kunin et al., 2019).

In this work, we aim to advance the theoretical understanding of linear autoencoder (LAE) models' generalizability using statistical learning theory. While generalization theory has been extensively studied for various machine learning and deep learning models (Vapnik, 1991; Dziugaite & Roy, 2017), its application to LAE recommendation models remains largely unexplored. To address this gap, we leverage PAC-Bayes theory (McAllester, 1998), which integrates the Probably Approximately Correct (PAC) framework with Bayesian inference. Our analysis produces a nonvacuous bound, offering practical insights into LAE performance on unseen data.

Our study to establish PAC-Bayes bounds for LAE models builds on the theoretical framework introduced by Shalaeva (Shalaeva et al., 2020), which provides a PAC-Bayes bound for multiple linear regression (a single dependent variable with multiple independent variables) under the assumption of Gaussian data. However, applying this framework to LAE models introduces several challenges:

1. **Multivariate Linear Regression**: The PAC-Bayes bound must be extended from the *multiple linear regression* setting to the *multivariate linear regression* scenario, which involves multiple dependent variables. Notably, PAC-Bayes bounds for multivariate linear regression – an important method and topic in statistical learning and inference – remain unexplored in the existing literature.

2. **Additional Convergence Requirements**: Our analysis reveals the need for additional convergence conditions beyond those presented in (Shalaeva et al., 2020). These conditions are essential for ensuring theoretical convergence in the more complex multivariate setting.

---

[1]Anonymous Institution, Anonymous City, Anonymous Region, Anonymous Country. Correspondence to: Anonymous Author <anon.email@domain.com>.

Preliminary work. Under review by the International Conference on Machine Learning (ICML). Do not distribute.

3. **Constraints on the Weight Matrix and Data**: LAE models impose a structural zero-diagonal constraint on the weight matrix and a data dependency constraint between the input and target. These constraints present unique challenges in theoretical analysis and practical calculation when adapting PAC-Bayes bounds from multivariate linear regression to LAE models.

This paper addresses the aforementioned challenges and makes the following key contributions:

- (Section 3) We develop a general theoretical PAC-Bayes bound for multivariate linear regression (Theorem 3.2), of which Shalaeva's bound (Shalaeva et al., 2020) for multiple linear regression is a special case. Additionally, we propose sufficient conditions (Theorem 3.3) that guarantee convergence for both the new bound (Theorem 3.2) and Shalaeva's original bound (Shalaeva et al., 2020).

- (Section 4) We adapt Theorem 3.2 to bounded data assumption (Assumption 4.1) and propose the PAC-Bayes bound for multivariate linear regression on bounded data (Section 4.1). We then show the bound for LAEs is a special case of this, achieved by applying the zero diagonal constraint and the data dependent constraint (Section 4.2).

- (Section 5) We propose practical methods for calculating and minimizing the PAC-Bayes Bound for LAEs, including solving for the optimal posterior distribution (Theorem 5.2) under Gaussian assumption (Assumption 5.1), and addressing the high computational complexity introduced by the zero diagonal constraint with a practical upper bound (Theorem 5.4).

- (Section 6) We evaluate the bound for LAEs on a specific data distribution (which is typically considered unknown in practice but assumed to be known in this experiment) to address the incomputability issue of the oracle bound, and the results show that our bound is non-vacuous.

All proofs of the theorems, lemmas and propositions presented in this paper are provided in Appendix A, Related Works are discussed in Appendix C, and Discussions are presented in Appendix E.

## 2. Preliminaries

**Alquier's Bound** (Alquier et al., 2016): Let $S = \{(x_i, y_i)\}_{i=1}^m$ be the dataset where $x_i \in \mathbb{R}^n$ is the feature vector and $y_i \in \mathbb{R}$ is the label. Suppose each $(x_i, y_i)$ is i.i.d. sampled from an unknown data distribution $\mathcal{D}$. Let $f_\theta : \mathbb{R}^n \to \mathbb{R}$ be the machine learning model where $\theta$ is the vector of parameters. Let $l$ be the loss function, $R^{\text{emp}}(\theta) = \frac{1}{m} \sum_{i=1}^m l(f_\theta(x_i), y_i)$ be the empirical risk and $R^{\text{true}}(\theta) = \mathbb{E}_{(x,y)\sim\mathcal{D}}[l(f_\theta(x), y)]$ be the true risk. Let $\pi$ be a prior distribution of $\theta$ and $\rho$ be the posterior distribution

of $\theta$, then for any $\lambda > 0, \delta > 0$,

$$P\left(\mathbb{E}_{\theta\sim\rho}[R^{\text{true}}(\theta)] < \mathbb{E}_{\theta\sim\rho}[R^{\text{emp}}(\theta)] + \frac{1}{\lambda}\left[D(\rho \,\|\, \pi) + \ln\frac{1}{\delta} + \Psi_{\pi,\mathcal{D},l}(\lambda, m)\right]\right) \geq 1 - \delta$$

where $\Psi_{\pi,\mathcal{D},l}(\lambda, m) = \ln \mathbb{E}_{\theta\sim\pi}\mathbb{E}_{S\sim\mathcal{D}^m}[e^{\lambda(R^{\text{true}}(\theta) - R^{\text{emp}}(\theta))}]$.

Note that the Alquier's bound involves $R^{\text{true}}(W)$ on both the left and right hand sides, meaning the bound cannot be computed without the knowledge of the data distribution $\mathcal{D}$. Such bound is refered to as the *oracle bound* (Alquier, 2021).

**Shalaeva's Bound** (Shalaeva et al., 2020): In Alquier's bound, suppose $f_\theta(x) = \theta^T x$ where $\theta \in \mathbb{R}^n$. Assume $\mathcal{D}$ satisfies $x_i \sim \mathcal{N}(0, \sigma_x^2 I)$, and there exist $\theta^* \in \mathbb{R}^n$ such that $y_i = (\theta^*)^T x_i + e_i$ where $e_i \sim \mathcal{N}(0, \sigma_e^2)$. Here $\sigma_x^2, \sigma_e^2$ are constants. Let the loss function be $l(f_\theta(x_i), y_i) = (\theta^T x_i - y_i)^2$, then

$$\Psi_{\pi,\mathcal{D},l}(\lambda, m) = \ln \mathbb{E}_{\theta\sim\pi} \frac{\exp(\lambda v_\theta)}{(1 + \frac{\lambda v_\theta}{m/2})^{m/2}} \leq \ln \mathbb{E}_{\theta\sim\pi} \exp\left(\frac{2\lambda^2 v_\theta^2}{m}\right) \tag{1}$$

where $v_\theta = \sigma_x^2 \|\theta - \theta^*\|_2^2 + \sigma_e^2$.

**Convergence of Shalaeva's Bound**: The convergence analysis in (Shalaeva et al., 2020) is presented informally. Here we formally state their results as follows:

(1) Since $\lim_{m\to\infty}(1 + \frac{\lambda v_\theta}{m/2})^{m/2} = \exp(\lambda v_\theta)$, for any $\lambda > 0$, the term $\Psi_{\pi,\mathcal{D},l}(\lambda, m)$ converges,

$$\lim_{m\to\infty} \Psi_{\pi,\mathcal{D},l}(\lambda, m) = \lim_{m\to\infty} \ln \mathbb{E}_{\theta\sim\pi} \frac{\exp(\lambda v_\theta)}{(1 + \frac{\lambda v_\theta}{m/2})^{m/2}}$$

$$= \ln \mathbb{E}_{\theta\sim\pi} \lim_{m\to\infty} \frac{\exp(\lambda v_\theta)}{(1 + \frac{\lambda v_\theta}{m/2})^{m/2}} = 0$$

(2) Let $d$ be a constant and $\lambda = m^{1/d}$, then $\ln \mathbb{E}_{\theta\sim\pi} \exp\left(\frac{2\lambda^2 v_\theta^2}{m}\right) = \ln \mathbb{E}_{\theta\sim\pi} \exp\left(2m^{2/d-1} v_\theta^2\right)$.

When $d > 2, \lim_{m\to\infty} m^{-1/d} \ln \mathbb{E}_{\theta\sim\pi} \exp\left(2m^{2/d-1} v_\theta^2\right) = 0$, thus the entire bound converges as $m \to \infty$.

$$\lim_{m\to\infty} \frac{1}{\lambda}\left[D(\rho \,\|\, \pi) + \ln\frac{1}{\delta} + \Psi_{\pi,\mathcal{D},l}(\lambda, m)\right] \leq$$

$$\lim_{m\to\infty} m^{-1/d}\left[D(\rho \,\|\, \pi) + \ln\frac{1}{\delta}\right] +$$

$$\lim_{m\to\infty} m^{-1/d} \ln \mathbb{E}_{\theta\sim\pi} \exp\left(2m^{2/d-1} v_\theta^2\right) = 0$$

Upon careful examination of their analysis, we found that additional conditions are needed to ensure the above convergency results, which were not discussed in their original paper. In (1), swapping $\lim$ and $\mathbb{E}$ is valid only under some specific conditions. For example, by *dominated convergence theorem* (Resnick, 1998; Rudin, 1976), the condition

can be $\mathbb{E}_{\theta \sim \pi}[\exp(\lambda v_\theta)] < \infty$. $\pi$ needs to be a distribution satisfying this condition. In (2), some choices of $\pi$ can cause divergence. For example, when $\pi$ is Gaussian distribution, we have $\ln \mathbb{E}_{\theta \sim \pi} \exp\left(2m^{2/d-1} v_\theta^2\right) = \infty$ for any $m > 0$, thus $\lim_{m \to \infty} m^{-1/d} \ln \mathbb{E}_{\theta \sim \pi} \exp\left(2m^{2/d-1} v_\theta^2\right) = \infty$ and the bound diverges. We will discuss these issues in Section 3.2.

**Multivariate Linear Regression** (Johnson & Wichern, 2007): Let $S = \{(x_i, y_i)\}_{i=1}^m$ be the dataset where $x_i \in \mathbb{R}^n$ and $y_i \in \mathbb{R}^p$. Let $X = [x_1, x_2, ..., x_m] \in \mathbb{R}^{n \times m}$ be the input matrix, $Y = [y_1, y_2, ..., y_m] \in \mathbb{R}^{p \times m}$ be the target, $W \in \mathbb{R}^{p \times n}$ be the weight matrix of the linear model and $E = [e_1, e_2, ..., e_m] \in \mathbb{R}^{p \times m}$ be the error matrix. The linear regression is defined as

$$Y = WX + E$$

Usually we let the first dimension of every $x_i$ be 1, i.e., $X_{1*}$ is a vector of all 1s. We say the linear regression is *multivariate* if $p > 1$, and is *multiple* if $n > 2$.

We can apply a statistical assumption to the multivariate linear regression, where it is typically assumed that the errors $e_i$ and $e_j$ are independent for $i \neq j$, but the dimensions of each $e_i$ can be dependent. A common statistical assumption is shown in Assumption 3.1.

**LAE Model and Recommender System**: In a recommender system, let $x \in \{0,1\}^n$ be a user vector that $x_i = 1$ indicates the user has interacted with item $i$, and $x_i = 0$ indicates the user has not yet interacted with item $i$, but may potentially be interested in it. An LAE model is represented by a matrix $W \in \mathbb{R}^{n \times n}$, which takes $x$ as input and generates a prediction by $\hat{y} = Wx$. The prediction $\hat{y}$ fills in the 0s in $x$. If $x_i = 0$ and $\hat{y}_i$ is closer to 1, it suggests that the user is likely to be interested in item $i$, and the item will be recommended. Items with $\hat{y}_i$ closer to 0 will not be recommended. If $x_i = 1$, $\hat{y}_i$ is should ideally be close to 0, as the system should avoid recommending items that the user already knows.

Let $y \in \{0,1\}^n$ be the target vector used in evaluation. We consider the item $i$ with $x_i = 1$ and $y_i = 1$ *wrongly labeled*, as it indicates that the recommender system would suggest a redundant item the user already knows. Wrongly labeled items should be excluded from the evaluation, as they misrepresent the model's performance.

**EASE** (Steck, 2019): EASE is one of the most popular method for training LAE models (Jin et al., 2021). Let $R^{n \times m}$ be the data matrix and $W \in \mathbb{R}^{n \times n}$ be the weight matrix, then EASE obtains the LAE model $W$ by solving the following problem

$$\min_W \|R - WR\|_F^2 + \gamma \|W\|_F^2 \quad \text{s.t. } \text{diag}(W) = 0 \quad (2)$$

where $\gamma$ is the regularization parameter. Let $W_0$ be the

solution of Eq (2), then $W_0$ has closed from: Let $P = \left(RR^T + \gamma I\right)^{-1}$, then $(W_0)_{ij} = 0$ if $i = j$ and $(W_0)_{ji} = -P_{ij}/P_{jj}$ if $i \neq j$.

# 3. PAC-Bayes Bound for Multivariate Linear Regression

## 3.1. The Statistical Assumption and the Bound

**Assumption 3.1.** Suppose each $(x_i, y_i)$ in $S$ is i.i.d. sampled from a distribution $\mathcal{D}$. $\mathcal{D}$ is defined as: (1) $x_i \sim \mathcal{N}(\mu_x, \Sigma_x)$; (2) there exist $W^* \in \mathbb{R}^{p \times n}$ and $e \sim \mathcal{N}(0, \Sigma_e)$ such that for any given $x_i$, $y_i = W^* x_i + e$, in other words, $y_i | x_i \sim \mathcal{N}(W^* x_i, \Sigma_e)$. Here $\mu_x \in \mathbb{R}^n$, $\Sigma_x \in \mathbb{R}^{n \times n}$ is positive semi-definite, and $\Sigma_e \in \mathbb{R}^{p \times p}$ is positive-definite.

The positive semi-definite assumption of $\Sigma_x$ allows $\Sigma_x$ to be singular, implying that the Gaussian distribution is degenerate, i.e., its support is on a lower dimensional manifold embedded in $\mathbb{R}^n$. This includes the case that $x_i$ has its first dimension to be constant 1 and the other $n - 1$ dimensions to be Gaussian random variables. In this case, the first row and first column of $\Sigma_x$ are 0.

Let $W \in \mathbb{R}^{p \times n}$ be the weight matrix of the linear model, then the prediction of the model on $x_i$ is given by $\hat{y}_i = W x_i$. The error is $y_i - \hat{y}_i = (W^* - W)x_i + e \sim \mathcal{N}(\mu_W, \Sigma_W)$, where

$$\mu_W = \mathbb{E}[(W^* - W)x_i + e] = (W^* - W)\mathbb{E}[x_i] + \mathbb{E}[e] = (W^* - W)\mu_x$$

$$\Sigma_W = \mathbb{E}[(W^* - W)(x_i - \mu_x) + e)][(W^* - W)(x_i - \mu_x) + e]^T$$
$$= (W^* - W)\Sigma_x(W^* - W)^T + \Sigma_e$$

It is easy to verify that $\Sigma_W$ is positive-definite. Thus, $\Sigma_W$ has an eigenvalue decomposition $\Sigma_W = S^T \Lambda S$ where $S$ is orthogonal, $\Lambda = \text{diag}(\eta_1, \eta_2, ..., \eta_p)$ and $\eta_i > 0$ for all $i$. Note that $S$ and $\Lambda$ depend on $W$.

Define the loss of the sample $(x_i, y_i)$ as $\|y_i - Wx_i\|_F^2$, the empirical risk as $R^{\text{emp}}(W) = \frac{1}{m} \sum_{i=1}^m \|y_i - Wx_i\|_F^2$ and the true risk as $R^{\text{true}}(W) = \mathbb{E}_{(x,y) \sim \mathcal{D}}[\|y - Wx\|_F^2]$. Then we have the following bound:

**Theorem 3.2.** *Let $\pi$ be the prior distribution of $W$, $\rho$ be the posterior distribution of $W$. Denote $b = S\Sigma_W^{-1/2}\mu_W$. Then for any $\lambda > 0$ and $\delta > 0$,*

$$P\left(\mathbb{E}_{W \sim \rho}[R^{\text{true}}(W)] < \mathbb{E}_{W \sim \rho}[R^{\text{emp}}(W)] + \right.$$
$$\left. \frac{1}{\lambda}\left[D(\rho \| \pi) + \ln\frac{1}{\delta} + \Psi_{\pi, \mathcal{D}}(\lambda, m)\right]\right) \geq 1 - \delta \quad (3)$$

*where*
$$\Psi_{\pi, \mathcal{D}}(\lambda, m)$$

$$= \ln \mathbb{E}_{W \sim \pi}\left[\exp\left(\lambda\left(\text{tr}(\Sigma_W) + \mu_W^T \mu_W\right)\right) \frac{\exp\left(\sum_{i=1}^p \frac{-\lambda m b_i^2 \eta_i}{m + 2\lambda \eta_i}\right)}{\prod_{i=1}^p (1 + 2\lambda \eta_i/m)^{m/2}}\right]$$

$$\leq \ln \mathbb{E}_{W \sim \pi} \exp\left(\frac{2\lambda^2 \|\Sigma_W\|_F^2}{m}\right)$$

The bound of Theorem 3.2 is a general case of Shalaeva's bound. It can be reduced to Shalaeva's bound by taking $p = 1$, $\mu_x = 0$, $\Sigma_x = \sigma_x^2 I$ and $\Sigma_e = \sigma_e^2$ for some $\sigma_x, \sigma_e$.

### 3.2. Convergence Analysis

This section presents the convergence analysis of Theorem 3.2. We outline sufficient conditions that ensure convergence, thereby completing and rigorously formalizing the convergence analysis of Shalaeva's bound (Shalaeva et al., 2020)

We first discuss the convergence of $\Psi_{\pi,\mathcal{D}}(\lambda, m)$ term, then the entire bound. Theorem 3.3 gives a sufficient condition for the convergence of $\Psi_{\pi,\mathcal{D}}(\lambda, m)$ based on the dominated convergence theorem.

**Theorem 3.3.** *If $\lambda$ and $\pi$ satisfies $\mathbb{E}_{W \sim \pi} \left[ \exp \left( \lambda \| (\Sigma_x + \mu_x \mu_x^T)^{1/2} (W^* - W) \|_F^2 \right) \right] < \infty$, then $\lim_{m \to \infty} \Psi_{\pi,\mathcal{D}}(\lambda, m) = 0$.*

By Theorem 3.3, we can derive some special cases that make $\Psi_{\pi,\mathcal{D}}(\lambda, m)$ converge:

(1) If $\pi$ is a bounded distribution such that $\|W\|_F < G$ where $G$ is a constant, then for any $\lambda > 0$,

$$\mathbb{E}_{W \sim \pi} \left[ \exp \left( \lambda \| (\Sigma_x + \mu_x \mu_x^T)^{1/2} (W^* - W) \|_F^2 \right) \right]$$
$$\leq \mathbb{E}_{W \sim \pi} \left[ \exp \left( \lambda \| (\Sigma_x + \mu_x \mu_x^T)^{1/2} \|_F^2 \| W^* - W \|_F^2 \right) \right]$$
$$\leq \exp \left( \lambda \| (\Sigma_x + \mu_x \mu_x^T)^{1/2} \|_F^2 \left( \| W^* \|_F + \| W \|_F \right)^2 \right)$$
$$< \exp \left( \lambda \| (\Sigma_x + \mu_x \mu_x^T)^{1/2} \|_F^2 \left( \| W^* \|_F + G \right)^2 \right) < \infty$$

(2) If $\pi$ is a distribution that for $W \sim \pi$, each $W_{ij}$ is independently sampled from $\mathcal{N}((\mathcal{U}_0)_{ij}, \sigma^2)$ where $\sigma > 0$ is a constant and $\mathcal{U}_0 \in \mathbb{R}^{n \times n}$. Then for any $\lambda \in (0, \frac{1}{2\eta_1 \sigma^2})$, $\mathbb{E}_{W \sim \pi} \left[ \exp \left( \lambda \| (\Sigma_x + \mu_x \mu_x^T)^{1/2} (W^* - W) \|_F^2 \right) \right] < \infty$ holds. This is because, let $\Sigma_x + \mu_x \mu_x^T = S^T \Lambda S$ be the eigenvalue decomposition and suppose $\Lambda = \text{diag}(\eta_1, \eta_2, ..., \eta_n)$ where $\eta_1$ is the largest eigenvalue, then

$$\mathbb{E}_{W \sim \pi} \left[ \exp \left( \lambda \| (\Sigma_x + \mu_x \mu_x^T)^{1/2} (W^* - W) \|_F^2 \right) \right]$$
$$= \prod_{i=1}^{p} \prod_{j=1}^{p} \frac{\exp \left( \frac{\lambda \eta_j \left( S_{j*} (W^* - \mathcal{U}_0)_{*i} \right)^2}{1 - 2\lambda \sigma^2 \eta_j} \right)}{(1 - 2\lambda \sigma^2 \eta_j)^{1/2}}$$

And $\lambda \in (0, \frac{1}{2\eta_1 \sigma^2})$ ensures denominator $(1 - 2\lambda \sigma^2 \eta_j)^{1/2}$ is not zero or undefined for any $j$.

Now we discuss the convergence of the entire bound when $\lambda = m^{1/d}$. Since $\frac{1}{\lambda} \left[ D(\rho \| \pi) + \ln \frac{1}{\delta} \right]$ surely converges as $m \to \infty$, we only discuss the convergence of $\frac{1}{\lambda} \Psi_{\pi,\mathcal{D}}(\lambda, m)$. By Theorem 3.2, $\frac{1}{\lambda} \Psi_{\pi,\mathcal{D}}(\lambda, m)$ converges if the upper bound $\frac{1}{\lambda} \ln \mathbb{E}_{W \sim \pi} \exp \left( \frac{2\lambda^2 \|\Sigma_W\|_F^2}{m} \right)$ converges.

(3) If $\pi$ is a bounded distribution satisfying $\|W\|_F < G$,

then

$$\|\Sigma_W\|_F^2 = \| (W^* - W) \Sigma_x (W^* - W)^T + \Sigma_e \|_F^2$$
$$\leq \left( \| (W^* - W) \Sigma_x (W^* - W)^T \|_F + \| \Sigma_e \|_F \right)^2$$
$$\leq \left( \| \Sigma_x \|_F \| W^* - W \|_F^2 + \| \Sigma_e \|_F \right)^2$$
$$\leq \left( \| \Sigma_x \|_F \left( \| W^* \|_F + \| W \|_F \right)^2 + \| \Sigma_e \|_F \right)^2$$
$$< \left( \| \Sigma_x \|_F \left( \| W^* \|_F + G \right)^2 + \| \Sigma_e \|_F \right)^2 < \infty$$

Denote $G' = \left( \| \Sigma_x \|_F \left( \| W^* \|_F + G \right)^2 + \| \Sigma_e \|_F \right)^2$. The upper bound converges when $d > 2$:

$$\lim_{m \to \infty} m^{-1/d} \ln \mathbb{E}_{W \sim \pi} \exp \left( 2m^{2/d-1} \| \Sigma_W \|_F^2 \right)$$
$$< \lim_{m \to \infty} m^{-1/d} \ln \mathbb{E}_{W \sim \pi} \exp \left( 2m^{2/d-1} G' \right) = 0$$

(4) If $\pi$ is a distribution that for $W \sim \pi$, each $W_{ij}$ is a Gaussian random variable, then the upper bound diverges when $d > 2$, thus we cannot show the convergence of $\frac{1}{\lambda} \Psi_{\pi,\mathcal{D}}(\lambda, m)$. We prove the divergence of the upper bound as follows. First, for any $r, q \in \{1, 2, ..., p\}$,

$$\| \Sigma_W \|_F^2 = \sum_{i=1}^{p} \sum_{j=1}^{p} \left( (W^* - W)_{*i}^T \Sigma (W^* - W)_{*j} + (\Sigma_e)_{ij} \right)^2$$
$$\geq \left( (W^* - W)_{*q}^T \Sigma_x (W^* - W)_{*q} + (\Sigma_e)_{qq} \right)^2$$
$$= \left( \| (\Sigma_x)^{1/2} (W^* - W)_{*q} \|_2^2 + (\Sigma_e)_{qq} \right)^2$$
$$\geq \left( \| (\Sigma_x)^{1/2} (W^* - W)_{*q} \|_2^2 \right)^2 \geq \left( (\Sigma_x)_{r*}^{1/2} (W^* - W)_{*q} \right)^4$$

In the above inequality we use the fact that $(\Sigma_x)_{qq} \geq 0$ since it is a diagonal element of $\Sigma_x$. Since $(W^* - W)_{*q}$ is a random Gaussian vector, $(\Sigma)_{r*}^{1/2} (W^* - W)_{*q}$ is a Gaussian random variable. Denote $w = (\Sigma)_{r*}^{1/2} (W^* - W)_{*q}$, then

$$m^{-1/d} \ln \mathbb{E}_{W \sim \pi} \exp \left( 2m^{2/d-1} \| \Sigma_W \|_F^2 \right) \geq m^{-1/d} \ln \mathbb{E}_w \exp \left( 2m^{2/d-1} w^4 \right)$$

**Lemma 3.4.** *Let $\{a_k\}_{i=0}^k$ be a sequence of real numbers. Let $X$ be a Gaussian random variable and $Y_k = \sum_{i=0}^k a_i X^i$ where $a_k > 0$. If $k \geq 3$, then $Y_k$ has no MGF, i.e., $M_{Y_k}(t) = \mathbb{E}_{Y_k}[\exp(tY_k)] = \mathbb{E}_X[\exp(tY_k)] = \infty$ for any $t > 0$.*

Lemma 3.4 states that any polynomial of Gaussian random variables of degree $\geq 3$ has no MGF. The term $w^4$ satisfies the conditions of Lemma 3.4 as a polynomial of degree 4. Thus we have $\mathbb{E}_w \exp \left( 2m^{2/d-1} w^4 \right) = \infty$ for any $m > 0$, and $\ln \mathbb{E}_w \exp \left( 2m^{2/d-1} w^4 \right) = \infty$. Note that when $m \to \infty$, $m^{-1/d}$ and $m^{2/d-1}$ are positive numbers being arbitrary close to 0 but never equivalent to 0. Thus $\lim_{m \to \infty} m^{-1/d} \ln \mathbb{E}_w \exp \left( 2m^{2/d-1} w^4 \right) = \infty$. This shows the upper bound diverges.

Recall that Shalaeva's bound in Section 2 has $v_\theta = \sigma_x^2 \| \theta - \theta^* \|_2^2 + \sigma_e^2$. When $\theta$ is a Gaussian vector, $v_\theta^2$ becomes

a polynomial of Gaussian random variables of degree 4, which satisfies the condition of Lemma 3.4. Thus the divergence $\lim_{m \to \infty} \ln \mathbb{E}_{\theta \sim \pi} \exp\left(2m^{2/d-1} v_\theta^2\right) = \infty$ cannot be resolved by taking any $d > 2$.

## 4. PAC-Bayes Bound for LAEs

This section demonstrates that, in the testing stage, the LAE model with squared loss can be viewed as a special case of multivariate linear regression on bounded data. We first propose a PAC-Bayes bound for multivariate linear regression on bounded data, which is derived by adjusting the data distribution assumption in Theorem 3.2 from Gaussian to one with bounded support (Section 4.1). We then apply the bound to LAEs (Section 4.2).

### 4.1. Adjusting the PAC-Bayes Bound to Bounded Data Assumption

Most real-world recommendation datasets are not Gaussian but bounded. For example, the dataset can be a binary matrix $R \in \{0,1\}^{n \times m}$ where $R_{ij} = 1$ means that user $j$ has interacted with item $i$, and $R_{ij} = 0$ means user $j$ has not interacted with item $i$. To apply the PAC-Bayes bound of Theorem 3.2 to recommendation datasets, we need to change the Gaussian data assumption (Assumption 3.1) to bounded data assumption.

We first consider the general definitions of empirical and true risk. Let $X \in \mathbb{R}^{n \times m}$ be the input, $Y \in \mathbb{R}^{n \times m}$ be the target, and $W \in \mathbb{R}^{n \times n}$ be the weight matrix. The empirical risk is

$$R^{\text{emp}}(W) = \frac{1}{m} ||Y - WX||_F^2 \tag{4}$$

Assume each pair $(X_{*j}, Y_{*j})$ is i.i.d. sampled from a $2n$ dimensional distribution $\mathcal{D}$, then we can define the true risk as

$$R^{\text{true}}(W) = \mathbb{E}_{(x,y) \sim \mathcal{D}}\left[||y - Wx||_F^2\right] \tag{5}$$

Now, we introduce the bounded data assumption, which assumes $\mathcal{D}$ is a distribution with bounded support:

**Assumption 4.1.** Suppose $\mathcal{D}$ is characterized by three finite cross-correlation matrices $\Sigma_{xx} = \mathbb{E}_{(x,y) \sim \mathcal{D}}[xx^T]$, $\Sigma_{xy} = \mathbb{E}_{(x,y) \sim \mathcal{D}}[xy^T]$ and $\Sigma_{yy} = \mathbb{E}_{(x,y) \sim \mathcal{D}}[yy^T]$, and $\Sigma_{xx}$ is positive definite.

**Lemma 4.2.** *Under Assumption 4.1, given any $W$, the true risk in Eq (5) can be expressed as*

$$R^{\text{true}}(W) = ||W\Sigma_{xx}^{1/2} - \Sigma_{xy}^T \Sigma_{xx}^{-1/2}||_F^2 - ||\Sigma_{xy}^T \Sigma_{xx}^{-1/2}||_F^2$$
$$+ \text{tr}(\Sigma_{yy}) \tag{6}$$

Then the PAC-Bayes bound for multivariate linear regression on bounded data is as follows (the same form as Eq (2) but with different settings):

$$P\left(\mathbb{E}_{W \sim \rho}[R^{\text{true}}(W)] < \mathbb{E}_{W \sim \rho}[R^{\text{emp}}(W)] + \frac{1}{\lambda} D(\rho \,||\, \pi)\right.$$
$$\left. + \frac{1}{\lambda} \ln \frac{1}{\delta} + \frac{1}{\lambda} \Psi_{\pi,\mathcal{D}}(\lambda, m)\right) \geq 1 - \delta \tag{7}$$

with $R^{\text{emp}}(W)$ given by Eq (4), $R^{\text{true}}(W)$ given by Eq (6), $\Psi_{\pi,\mathcal{D}}(\lambda, m) = \ln \mathbb{E}_{W \sim \pi} \mathbb{E}_{R \sim \mathcal{D}^m}\left[e^{\lambda(R^{\text{true}}(W) - R^{\text{emp}}(W))}\right]$.

### 4.2. Applying the PAC-Bayes Bound to LAEs

Generally, an LAE model is represented by a weight matrix $W \in \mathbb{R}^{n \times n}$, with the zero diagonal constraint $\text{diag}(W) = 0$ being optionally applied, depending on the method used to obtain $W$. For example, if $W$ is obtained through training with EASE or EDLAE, the zero diagonal constraint is applied.

We consider training and evaluation as independent stages and focus only on evaluation. Suppose we have obtained an LAE model $W$ through any method, whether trained via EASE or EDLAE, or random initialized but untrained. Let $R \in \{0,1\}^{n \times m}$ be the matrix used for testing. To avoid the wrongly labeled items as mentioned in Section 2, we split the 1s in $R$ into two matrices $X \in \{0,1\}^{n \times m}$ and $Y \in \{0,1\}^{n \times m}$ in the following way: For any $i \in \{1,2,...,n\}$ and $j \in \{1,2,...,m\}$, if $R_{ij} = 0$, we set $X_{ij} = Y_{ij} = 0$; if $R_{ij} = 1$, we set $X_{ij} = 1, Y_{ij} = 0$ with probability $p$ and $X_{ij} = 0, Y_{ij} = 1$ with probability $1 - p$. In this case, $X_{ij}$ and $Y_{ij}$ cannot both be 1. For each pair $(X_{*j}, Y_{*j})$, the prediction of LAE model is $WX_{*j}$ and the error vector is $Y_{*j} - WX_{*j}$.

Therefore, when using the squared loss (squared Frobenius norm) as the evaluation metric, the LAE model can be considered a special case of multivariate linear regression on bounded data with two constraints:

(1) Zero diagonal constraint on $W$: $\text{diag}(W) = 0$ (Optional).

(2) Data dependent constraint on $X$ and $Y$: For any $i, j$, $X_{ij}$ and $Y_{ij}$ are either 0 or 1, but cannot both be 1.

Now we apply the data dependent constraint of LAE to Assumption 4.1. Since $X$ and $Y$ are generated from $R$, we make the following statistical assumptions on $R$: Let $\mathcal{M}$ be an $n$ dimensional multivariate Bernoulli distribution. Suppose each $R_{*j}$ is i.i.d. sampled from $\mathcal{M}$. Let $r \sim \mathcal{M}$ be a random vector. Given $p \in (0,1)$, let $\Delta \in \{0,1\}^n$ be a random Bernoulli vector conditioned on $r$ such that $P(\Delta_i = 1|r_i = 1) = p$, $P(\Delta_i = 0|r_i = 1) = 1 - p$ and $P(\Delta_i = 0|r_i = 0) = 1$ for any $i \in \{1,2,...,n\}$. Let $x = \Delta \odot r$ and $y = (\mathbf{1} - \Delta) \odot r$ where $\mathbf{1}$ is a vector of all 1. In this case, the random variables $(x, y)$ are represented by $(\Delta, r)$, and the true risk of Eq (5) is rewritten as

$$R^{\text{true}}(W) = \mathbb{E}_{\Delta, r}\left[||(\mathbf{1} - \Delta) \odot r - W(\Delta \odot r)||_F^2\right] \tag{8}$$

And Lemma 4.2 is applied to this case as follows:

**Lemma 4.3.** *Denote $\Sigma_{rr} = \mathbb{E}_{r \sim \mathcal{M}}[rr^T]$. Eq (8) can be written in the same form as Eq (6) by plugging in*

$$\Sigma_{xx} = p^2 \Sigma_{rr} + p(1-p)(I \odot \Sigma_{rr})$$
$$\Sigma_{yy} = (1-p)^2 \Sigma_{rr} + p(1-p)(I \odot \Sigma_{rr})$$
$$\Sigma_{xy} = p(1-p)(\Sigma_{rr} - I \odot \Sigma_{rr})$$

*Here $\odot$ means element-wise product. Also, $\Sigma_{xx}$ is positive definite if $\Sigma_{rr}$ is positive definite.*

## 5. A Practical Method for Calculating the LAE Bound

The right hand side of Eq (7) is as follows, where we highlight part 1 and part 2.

$$\underbrace{\mathbb{E}_{W \sim \rho}[R^{\text{emp}}(W)] + \frac{1}{\lambda}D(\rho \,\|\, \pi)}_{\text{part 1}} + \frac{1}{\lambda}\ln\frac{1}{\delta} + \underbrace{\frac{1}{\lambda}\Psi_{\pi,\mathcal{D}}(\lambda, m)}_{\text{part 2}} \quad (9)$$

For any given $\delta$, Eq (9) is a function of $\lambda, \pi, \rho$. Since Eq (7) holds for any $\lambda, \pi, \rho$, we aim to find a practical method for minimizing Eq (9) with respect to these parameters, and use the minimized bound to verify the non-vacuousness of Eq (7).

It is generally considered difficult to solve for $\lambda, \pi, \rho$ simultaneously (Alquier, 2021), so we typically solve for $\rho$ with $\lambda$ and $\pi$ fixed. We show how to minimize part 1 in Section 5.1 and how to find a practical upper bound for part 2 in Section 5.2.

### 5.1. Closed-form Solution for the Optimal $\rho$

Given $\pi$ and $\lambda$, we search for the optimal $\rho$ by

$$\min_{\rho} \mathbb{E}_{W \sim \rho}[R^{\text{emp}}(W)] + \frac{1}{\lambda}D(\rho \,\|\, \pi) \quad (10)$$

Usually we restrict $\pi$ and $\rho$ to be specific distributions that make Eq (10) easy to calculate. (Dziugaite & Roy, 2017) proposed a practical way to calculate the PAC-Bayes bound for deep neural networks, where they assumes $\pi$ and $\rho$ to be independent multivariate Gaussian. This enables the $D(\rho \,\|\, \pi)$ term to be easily calculated. We mainly follow the assumptions in (Dziugaite & Roy, 2017):

**Assumption 5.1.** Denote $\bar{\mathcal{N}}(\mathcal{A}, \mathcal{B})$ for some $\mathcal{A} \in \mathbb{R}^{n \times n}$ and non-negative $\mathcal{B} \in \mathbb{R}^{n \times n}$ as the multivariate Gaussian distribution that $W \sim \bar{\mathcal{N}}(\mathcal{A}, \mathcal{B})$ means $W \in \mathbb{R}^{n \times n}$ and each $W_{ij}$ is independently from $\mathcal{N}(\mathcal{A}_{ij}, \mathcal{B}_{ij})$. Assume $\rho$ is the distribution $\bar{\mathcal{N}}(\mathcal{U}, \mathcal{S})$ and $\pi$ is the distribution $\bar{\mathcal{N}}(\mathcal{U}_0, \sigma^2 J)$, where $\mathcal{U} \in \mathbb{R}^{n \times n}, \mathcal{U}_0 \in \mathbb{R}^{n \times n}, \mathcal{S} \in \mathbb{R}^{n \times n}$, $J = \{1\}^{n \times n}$ and $\sigma > 0$. $\mathcal{S}$ is a positive matrix if no constraint is applied.

Applying the constraint $\text{diag}(W) = 0$ to $\rho$ and $\pi$ is equivalent to set $\text{diag}(\mathcal{U}) = 0, \text{diag}(\mathcal{S}) = 0, \text{diag}(\mathcal{U}_0) = 0$ and $\text{diag}(\sigma^2 J) = 0$.

(Dziugaite & Roy, 2017) solved the optimal $\rho$ using stochastic gradient descent, where in each iteration the gradient is calculated by Monte Carlo method. It should be noticed that Dziugaite and Roy used the iterative method because they worked on the neural network model, for which the optimal $\rho$ may not have a closed-form solution. Due to the simplicity of LAE, we find that the optimal $\rho$ for Eq (10) has closed-form solution, as shown in Theorem 5.2 (1). This allows us to solve $\rho$ directly and avoid time-consuming iterative methods.

**Theorem 5.2.** *(1) Under Assumption 5.1, the closed-form solution of the optimal $\rho$ of Eq (10) is given by*

$$\mathcal{U} = \left(\frac{1}{m}YX^T + \frac{1}{2\lambda\sigma^2}\mathcal{U}_0\right)\left(\frac{1}{m}XX^T + \frac{1}{2\lambda\sigma^2}I\right)^{-1},$$

$$\mathcal{S}_{ij} = \frac{1}{\frac{2\lambda}{m}X_{j*}X_{j*}^T + \frac{1}{\sigma^2}} \quad \text{for } i,j \in \{1,2,...,n\}$$

*(2) If we add the constraint $\text{diag}(W) = 0$ to $\rho$ and $\pi$, then the optimal $\rho$ becomes*

$$\mathcal{S}_{ij} = \frac{1}{\frac{2\lambda}{m}X_{j*}X_{j*}^T + \frac{1}{\sigma^2}}, \ \mathcal{S}_{ii} = 0 \quad \text{for } i,j \in \{1,2,...,n\} \text{ and } i \neq j$$

$$\mathcal{U} = \left(\frac{1}{m}YX^T + \frac{1}{2\lambda\sigma^2}\mathcal{U}_0 - \frac{1}{2}\text{Diag}(x)\right)\left(\frac{1}{m}XX^T + \frac{1}{2\lambda\sigma^2}I\right)^{-1}$$

*where*

$$x = 2 \cdot \text{diag}\left[\left(\frac{1}{m}YX^T + \frac{1}{2\lambda\sigma^2}\mathcal{U}_0\right)\left(\frac{1}{m}XX^T + \frac{1}{2\lambda\sigma^2}I\right)^{-1}\right] \oslash$$

$$\text{diag}\left[\left(\frac{1}{m}XX^T + \frac{1}{2\lambda\sigma^2}I\right)^{-1}\right]$$

*Here $\oslash$ means element-wise division and $\text{Diag}(x)$ means expanding $x \in \mathbb{R}^n$ to an $n \times n$ diagonal matrix.*

### 5.2. Calculating $\Psi_{\pi,\mathcal{D}}(\lambda, m)$ under the Zero Diagonal Constraint

Since $\Psi_{\pi,\mathcal{D}}(\lambda, m) = \ln \mathbb{E}_{\pi}\mathbb{E}_{\mathcal{D}}[e^{\lambda(R^{\text{true}}(W) - R^{\text{emp}}(W))}]$ and $R^{\text{emp}}(W) \geq 0$, based on the idea of (Germain et al., 2016), we can get an upper bound of $\Psi$ by removing $-R^{\text{emp}}(W)$: Let $\Psi'_{\pi,\mathcal{D}}(\lambda) = \ln \mathbb{E}_{\pi}[e^{\lambda R^{\text{true}}(W)}]$, then $\Psi_{\pi,\mathcal{D}}(\lambda, m) \leq \Psi'_{\pi,\mathcal{D}}(\lambda)$. $\Psi'$ does not converge as $m \to \infty$ since it is independent of $m$, but it is easier to calculate than $\Psi$.

By Lemma 4.2, we have

$$\mathbb{E}_{\pi}\left[e^{\lambda R^{\text{true}}(W)}\right]$$

$$= \mathbb{E}_{\pi}\left[e^{\lambda\left(\|W\Sigma_{xx}^{1/2} - \Sigma_{xy}^T\Sigma_{xx}^{-1/2}\|_F^2 - \|\Sigma_{xy}^T\Sigma_{xx}^{-1/2}\|_F^2 + \text{tr}(\Sigma_{yy})\right)}\right]$$

$$= \mathbb{E}_{\pi}\left[e^{\lambda\|W\Sigma_{xx}^{1/2} - \Sigma_{xy}^T\Sigma_{xx}^{-1/2}\|_F^2}\right]e^{\lambda\left(\text{tr}(\Sigma_{yy}) - \|\Sigma_{xy}^T\Sigma_{xx}^{-1/2}\|_F^2\right)}$$

Let $B = -\Sigma_{xy}^T \Sigma_{xx}^{-1/2}$ and $C = e^{\lambda\left(\text{tr}(\Sigma_{yy}) - \|\Sigma_{xy}^T \Sigma_{xx}^{-1/2}\|_F^2\right)}$, then

$$\mathbb{E}_\pi \left[ e^{\lambda R^{\text{true}}(W)} \right] = C\, \mathbb{E}_\pi \left[ e^{\lambda\|W\Sigma_{xx}^{1/2} + B\|_F^2} \right]$$

$$= C\, \mathbb{E}_\pi \left[ e^{\lambda \sum_{i=1}^n \|W_{i*}\Sigma_{xx}^{1/2} + B_{i*}\|_F^2} \right] \quad (11)$$

We first consider the case without the constraint $\text{diag}(W) = 0$. Since we assume $\pi = \bar{\mathcal{N}}(\mathcal{U}_0, \sigma^2 J)$ in Assumption 5.1, $W_{i*}^T \sim \mathcal{N}((\mathcal{U}_0)_{i*}^T, \sigma^2 I)$, thus $(W_{i*}\Sigma_{xx}^{1/2} + B_{i*})^T = \Sigma_{xx}^{1/2}W_{i*}^T + B_{i*}^T \sim \mathcal{N}(\Sigma_{xx}^{1/2}(\mathcal{U}_0)_{i*}^T + B_{i*}^T, \sigma^2\Sigma_{xx})$.

**Proposition 5.3.** *Let $A = \sigma^2\Sigma_{xx}$, and $A = S^T\Lambda S$ be the eigenvalue decomposition where $S$ is orthogonal and $\Lambda = diag(\eta_1, \eta_2, ..., \eta_n)$. Denote $\mu^i = \Sigma_{xx}^{1/2}(\mathcal{U}_0)_{i*}^T + B_{i*}^T$. Then we can rewrite Eq (11) as*

$$\mathbb{E}_\pi \left[ e^{\lambda R^{\text{true}}(W)} \right] = C \prod_{i=1}^n \prod_{j=1}^n \frac{\exp\left( \frac{\lambda(\bar{b}_j^i)^2 \eta_j}{1 - 2\lambda\eta_j} \right)}{(1 - 2\lambda\eta_j)^{1/2}}$$

$$\text{where } \bar{b}^i = SA^{-1/2}\mu^i \quad (12)$$

Now we discuss the case that $\text{diag}(W) = 0$ is applied. Denote $\pi'$ as the distribution $\pi$ with the constraint $\text{diag}(W) = 0$, that is, for $W \sim \pi'$, $W_{ii} = 0$ for all $i$. Then $\pi' = \bar{\mathcal{N}}(\mathcal{U}_0, \sigma^2(J - I))$ where $\text{diag}(\mathcal{U}_0) = 0$, and $W_{i*}^T \sim \mathcal{N}\left((\mathcal{U}_0)_{i*}^T, \sigma^2(I - I^i)\right)$ where $I^i$ is a matrix with $I_{ii}^i = 1$ and other entries being 0. Therefore, $(W_{i*}\Sigma_{xx}^{1/2} + B_{i*})^T \sim \mathcal{N}\left(\Sigma_{xx}^{1/2}(\mathcal{U}_0)_{i*}^T + B_{i*}^T, \sigma^2(\Sigma_{xx} - (\Sigma_{xx}^{1/2})_{*i}(\Sigma_{xx}^{1/2})_{*i}^T)\right)$.

Denote $A^{(i)} = \sigma^2(\Sigma_{xx} - (\Sigma_{xx}^{1/2})_{*i}(\Sigma_{xx}^{1/2})_{*i}^T)$, then $A^{(i)}$ is singular and positive semi-definite. Let $A^{(i)} = S^{(i)T}\Lambda^{(i)}S^{(i)}$ be the eigenvalue decomposition where $S^{(i)}$ is orthogonal and $\Lambda^{(i)} = \text{diag}(\eta_1^{(i)}, \eta_2^{(i)}, ..., \eta_n^{(i)})$. Then

$$\mathbb{E}_{\pi'} \left[ e^{\lambda R^{\text{true}}(W)} \right] = C \prod_{i=1}^n \prod_{j=1}^n \frac{\exp\left( \frac{\lambda(b_j^{(i)})^2 \eta_j^{(i)}}{1 - 2\lambda\eta_j} \right)}{\left(1 - 2\lambda\eta_j^{(i)}\right)^{1/2}}$$

$$\text{where } b^{(i)} = S^{(i)}(A^{(i)})^{-1/2}\mu^i \quad (13)$$

The issue with Eq (13) is its high computational complexity: We need to calculate the eigenvalue decomposition for each $A^{(i)}$ in order to obtain $S^{(i)}$ and $\Lambda^{(i)}$. Since each eigenvalue decomposition costs $O(n^3)$, the computation of Eq (13) costs $O(n^4)$, which is impractical.

Since $\mathbb{E}_{\pi'} \left[ e^{\lambda R^{\text{true}}(W)} \right]$ is computationally difficult, we can instead compute an upper bound with lower complexity. The following theorem establishes the upper bound:

**Theorem 5.4.** *Suppose $\pi' = \bar{\mathcal{N}}(\mathcal{U}_0, \sigma^2(J - I))$ and $\pi = \bar{\mathcal{N}}(\mathcal{U}_0, \sigma^2 J)$, then $\mathbb{E}_{\pi'} \left[ e^{\lambda R^{\text{true}}(W)} \right] \leq \mathbb{E}_\pi \left[ e^{\lambda R^{\text{true}}(W)} \right]$ for any $\lambda \in \left(0, \frac{1}{2\eta_1}\right)$.*

Theorem 5.4 holds for any $\mathcal{U}_0$, including the special case where $\text{diag}(\mathcal{U}_0) = 0$ for both $\pi'$ and $\pi$. Note that $\mathbb{E}_\pi \left[ e^{\lambda R^{\text{true}}(W)} \right]$ is much easier to compute: We only need to calculate the eigenvalue decomposition of $A$, so Eq (12) costs $O(n^3)$.

To compute Eq (12), we need to know $\Sigma_{xx}$, $\Sigma_{xy}$ and $\Sigma_{yy}$. Under the LAE constraints, these three matrices are generated by $\Sigma_{rr}$ according to Lemma 4.3, so we only need to know $\Sigma_{rr}$. However, in practice, we cannot determine the exact value of $\Sigma_{rr}$ since $\Sigma_{rr} = \mathbb{E}_{r\sim\mathcal{M}}[rr^T]$, and the distribution $\mathcal{M}$ is usually unknown.

Not all PAC-Bayes bounds face this issue. PAC-Bayes bounds are classified into two types: empirical bounds and oracle bounds (Alquier, 2021). Empirical bounds, such as Seeger's bound (Langford & Seeger, 2001) used in (Dziugaite & Roy, 2017), can be computed without requiring knowledge of the data distribution. Our bound is based on Alquier's oracle bound (Alquier et al., 2016), which requires knowledge of the data distribution $\mathcal{M}$ to compute – an impossible task unless one is an oracle. Oracle bounds are primarily used for theoretical analysis, and in practice, we can only compute empirical approximations of them.

### 5.3. The Final Bound

The last step in completing the bound is to determine how to choose $\lambda$. According to (Alquier, 2021), we can search $\lambda$ over a finite grid $\Lambda = \{\lambda_1, \lambda_2, ..., \lambda_L\}$, as detailed in Appendix B. Let $L$ be the number of elements in $\Lambda$. Applying the grid search for $\lambda$ to Eq (7), we obtain the final bound: with probability $1 - \delta$,

$$\mathbb{E}_{W\sim\rho}[R^{\text{true}}(W)] \leq$$

$$\mathbb{E}_{W\sim\rho}[R^{\text{emp}}(W)] + \frac{1}{\lambda}\left[ D(\rho\,||\,\pi) + \ln\frac{L}{\delta} + \ln\mathbb{E}_\pi\left[ e^{\lambda R^{\text{true}}(W)} \right] \right]$$

$$(14)$$

Now we summarize the methods for calculating Eq (14) under LAE constraints. We will apply the zero diagonal constraint $\text{diag}(W) = 0$ by default, while the non-constraint case can be derived similarly.

Given $\lambda$ and $\pi = \bar{\mathcal{N}}(\mathcal{U}_0, \sigma^2 I)$, the optimal $\rho = \bar{\mathcal{N}}(\mathcal{U}, \mathcal{S})$ that minimizes the right hand side of Eq (14) is obtained by Theorem 5.2. Once $\rho$ is obtained, we can calculate $\mathbb{E}_{W\sim\rho}[R^{\text{emp}}(W)]$ by Eq (25), which can be simplified as

$$\mathbb{E}_{W\sim\rho}[R^{\text{emp}}(W)]$$

$$= \frac{1}{m}\|Y - \mathcal{U}X\|_F^2 + \frac{n-1}{m}\|\text{Diag}(\mathcal{S}_{1*})^{1/2}X\|_F^2 \quad (15)$$

Similarly, by Eq (6), $\mathbb{E}_{W\sim\rho}[R^{\text{true}}(W)]$ can be expressed as

$$\mathbb{E}_{W\sim\rho}[R^{\text{true}}(W)] = \|\Sigma_{xy}^T\Sigma_{xx}^{-1/2} - \mathcal{U}\Sigma_{xx}^{1/2}\|_F^2$$

$$+ (n-1)\|\text{Diag}(\mathcal{S}_{1*})^{1/2}\Sigma_{xx}^{1/2}\|_F^2 + \text{tr}(\Sigma_{yy}) - \|\Sigma_{xy}^T\Sigma_{xx}^{-1/2}\|_F^2$$

$$(16)$$

By Theorem 5.4, the term $\mathbb{E}_\pi\left[e^{\lambda R^{\text{true}}(W)}\right]$ in Eq (14) holds when the constraint $\text{diag}(W) = 0$ is applied to $\pi$.

The calculation process for the bound Eq (14) is summarized in Algorithm 1.

---

**Algorithm 1** Calculation of the PAC-Bayes bound for LAEs

    **Input:** $\Sigma_{rr}, p, \delta, \sigma, \Lambda = \{\lambda_1, \lambda_2, ..., \lambda_L\}, X, Y$, and the LAE model $W$ (with $\text{diag}(W) = 0$).
    Calculate $\Sigma_{xx}, \Sigma_{xy}, \Sigma_{yy}$ with $\Sigma_{rr}, p$ by Lemma 4.3.
    Set $\pi = \bar{\mathcal{N}}(W, \sigma^2 I)$ (i.e., $\mathcal{U}_0 = W$).
    Let $H = \{\}$ be a set to store the results.
    **for** each $\lambda_i$ **in** $\Lambda$:
        Calculate $\rho = \bar{\mathcal{N}}(\mathcal{U}, \mathcal{S})$ with $\pi, \lambda_i$ by Theorem 5.2 (2).
        Calculate $D(\rho\,\|\,\pi)$ with $\rho, \pi$ by Eq (33).
        Calculate $\mathbb{E}_{W\sim\rho}[R^{\text{emp}}(W)]$ with $\rho, X, Y$ by Eq (15).
        Calculate $\mathbb{E}_{W\sim\rho}[R^{\text{true}}(W)]$ with $\rho, \Sigma_{xx}, \Sigma_{xy}, \Sigma_{yy}$ by Eq (16).
        Calculate $\mathbb{E}_\pi\left[e^{\lambda R^{\text{true}}(W)}\right]$ with $\pi, \Sigma_{xx}, \Sigma_{xy}, \Sigma_{yy}, \lambda_i$ by Eq (12).
        Calculate the right hand side of Eq (14) and let the result be $\text{RH}_i$. Let $\text{LH}_i = \mathbb{E}_{W\sim\rho}[R^{\text{true}}(W)]$.
        Append $(\text{LH}_i, \text{RH}_i)$ to $H$.
    **Output:** the pair $(\text{LH}^*, \text{RH}^*)$ in $H$ that $\text{RH}^*$ is minimal.

---

## 6. Experiments

It is difficult to determine whether the bound Eq (14) is non-vacuous theoretically, especially since the value of $D(\rho\,\|\,\pi)$ unknown. So we conduct experiments on real world datasets to calculate its exact value.

The main idea of our experiment is as follows: Since Eq (14) is an oracle bound, it is impossible to calculate without knowing the data distribution $\mathcal{M}$. Let $\mathcal{M}^*$ be a special case of $\mathcal{M}$ such that $\Sigma_{rr} = \mathbb{E}_{r\sim\mathcal{M}^*}\left[rr^T\right] = \frac{1}{m}RR^T$, where $R \in \{0,1\}^{n\times m}$ is our dataset. We show that this bound is non-vacuous on $\mathcal{M}^*$.

We split the entire dataset $R^{n\times m}$ into a training set $R_{\text{train}}^{n\times m_1}$ and a test set $R_{\text{test}}^{n\times(m-m_1)}$ where we set $m_1 = 0.7m$. The test set $R_{\text{test}}$ is further split into $X$ and $Y$ by assigning each 1 in $R$ to $X$ with probability $p = \frac{1}{2}$ and to $Y$ with probability $1 - p = \frac{1}{2}$. The LAE model $W$ is obtained by solving the EASE target function Eq (2) using the data $R_{\text{train}}$ (The LAE model can also be obtained using other methods. We use EASE here as an example). We set $\gamma$ in Eq (2) to be $50, 100$ and $200$ to obtain three different LAE models and test them accordingly.

Our experiments run on a machine with 500 GB RAM and a Nvidia A100 GPU. The GPU has 80 GB RAM. We use three datasets: MovieLens 20M (ML 20M), Netflix and MSD,

with their details shown in Table 1. The computation of PAC-Bayes bound for LAE mainly follows Algorithm 1. The other parameters are set as follows: $\delta = 0.01$, $\sigma = 0.001$, $\Lambda = \{1, 2, 4, 8, 16, 32, 64, 128, 256, 512\}$.

*Table 1.* Dataset information

| Dataset | ML 20M | Netflix | MSD |
|---------|--------|---------|-----|
| #rows | 138493 | 480189 | 1017982 |
| #columns | 26744 | 17770 | 40000 |
| #ratings | 2000263 | 100480507 | 33687193 |

The results are presented in Table 2, where each pair $(\text{LH}, \text{RH})$ is the output of Algorithm 1. LH is the left hand side of Eq (14) while RH is the right hand side.

We evaluate the non-vacuousness by comparing the gap between LH and RH. To the best of our knowledge, there is no universally accepted definition for how small the gap must be to consider a theoretical bound non-vacuous. (Dziugaite & Roy, 2017) showed in their experiments that a bound with RH within 10 times LH (or some empirical estimation of LH) can be considered non-vacuous. We adopt this criterion in our work. Table 2 shows that RH is within 3 times LH in all cases, so our bound is non-vacuous. Additionally, the values of the terms in RH are presented in Table 3 in Appendix D, which shows that the value of $D(\rho\,\|\,\pi)$ is typically trivial.

*Table 2.* Experiment results of the PAC-Bayes bound for LAE

| Models | | ML 20M | Netflix | MSD |
|--------|------|--------|---------|-----|
| $\gamma = 50$ | LH | 61.66 | 87.22 | 15.96 |
| | RH | 128.66 | 178.11 | 32.60 |
| $\gamma = 100$ | LH | 60.75 | 86.54 | 15.85 |
| | RH | 125.90 | 176.25 | 32.26 |
| $\gamma = 200$ | LH | 60.06 | 85.96 | 15.76 |
| | RH | 123.67 | 174.55 | 31.94 |

## 7. Conclusions

This paper studies the generalizability of multivariate linear regression and LAEs. We propose a new PAC-Bayes bound for multivariate linear regression, which generalizes Shalaeva's bound for multiple linear regression (Shalaeva et al., 2020). We also present a convergence analysis and demonstrate the sufficient conditions that ensure the bound's convergence.

We extend the PAC-Bayes bound from multivariate linear regression to LAEs by demonstrating that an LAE with squared loss is a special case of multivariate linear regression on bounded data. We also propose practical methods for calculating the bound under the constraints introduced by LAEs, and the non-vacuousness of the bound is validated through experiments.

## Impact Statements

This work advances the theoretical foundations of machine learning by introducing the first PAC-Bayes bound for multivariate linear regression, extending beyond single-output regression to handle multiple dependent variables simultaneously. This establishes new generalization guarantees for structured prediction, multi-task learning, and recommendation systems. Additionally, we identify and correct a limitation in an existing PAC-Bayes proof for single-output linear regression, further strengthening the theoretical foundation of regression analysis.

Building on this, we apply our bound to linear autoencoders (LAEs) in recommendation systems, delivering their first rigorous generalization analysis. Our approach accounts for key structural constraints, such as the zero-diagonal weight requirement, ensuring applicability to models like EASE and EDLAE.

Beyond theory, our work has direct practical implications for model evaluation and selection. Our bound provides a post-training diagnostic tool for assessing the generalization of any LAE model—regardless of its training process (EASE, EDLAE, or random initialization). While not directly guiding training or hyperparameter tuning, a smaller PAC-Bayes bound suggests better generalization on unseen data. Empirical results confirm that our bound remains within a reasonable multiple of the test error, offering reliable probabilistic estimates of true risk independent of training error.

Our work focuses on theoretical generalization analysis and poses no immediate ethical risks. However, recommendation systems shape content exposure and user behavior in domains like e-commerce and social media. Strengthening generalization theory alongside other recommendation criteria may help mitigate bias, enhance fairness, and improve trust in AI-driven systems.

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

## A. Proofs of the Theorems

*Proof of Theorem 3.2*:

Given $W$, let $(x, y) \sim \mathcal{D}$, and denote $v = y - Wx$, then $v \sim \mathcal{N}(\mu_W, \Sigma_W)$. Suppose there exists $Q \in \mathbb{R}^{p \times p}$ such that $\Sigma_W = QQ^T$. Such $Q$ exists since we can take $Q = \Sigma_W^{1/2} = S^T \Lambda^{1/2} S$, but we do not assume it to be unique. Let $\epsilon \sim \mathcal{N}(0, I)$, then we can write $v = Q\epsilon + \mu_W$. Thus,

$$
\begin{aligned}
R^{\text{true}}(W) &= \mathbb{E}_{(x,y) \sim \mathcal{D}} \left[ \|y - Wx\|_F^2 \right] = \mathbb{E}_\epsilon \left[ \|Q\epsilon + \mu_W\|_F^2 \right] = \mathbb{E}_\epsilon \left[ (Q\epsilon + \mu_W)^T (Q\epsilon + \mu_W) \right] \\
&= \mathbb{E}_\epsilon [\epsilon^T Q^T Q \epsilon + \mu_W^T Q \epsilon + \epsilon^T Q^T \mu_W + \mu_W^T \mu_W] = \text{tr}(Q^T Q) + \mu_W^T \mu_W \\
&= \text{tr}(QQ^T) + \mu_W^T \mu_W = \text{tr}(\Sigma_W) + \mu_W^T \mu_W
\end{aligned}
\tag{17}
$$

Also, we can express the random variable $\|v\|_F^2$ in quadratic form (Representation 3.1a.1, (Mathai & Provost, 1992)):

$$
\begin{aligned}
\|v\|_F^2 = v^T v &= (Q\epsilon + \mu_W)^T (Q\epsilon + \mu_W) \\
&= (Q\epsilon + \mu_W)^T \Sigma_W^{-1/2} \Sigma_W \Sigma_W^{-1/2} (Q\epsilon + \mu_W) \\
&= (\Sigma_W^{-1/2} Q\epsilon + \Sigma_W^{-1/2} \mu_W)^T \Sigma_W (\Sigma_W^{-1/2} Q\epsilon + \Sigma_W^{-1/2} \mu_W) \\
&= (\Sigma_W^{-1/2} Q\epsilon + \Sigma_W^{-1/2} \mu_W)^T S^T \Lambda S (\Sigma_W^{-1/2} Q\epsilon + \Sigma_W^{-1/2} \mu_W) \\
&= (S\Sigma_W^{-1/2} Q\epsilon + S\Sigma_W^{-1/2} \mu_W)^T \Lambda (S\Sigma_W^{-1/2} Q\epsilon + S\Sigma_W^{-1/2} \mu_W)
\end{aligned}
$$

Denote $\epsilon' = S\Sigma_W^{-1/2} Q\epsilon$, then $\epsilon' \sim \mathcal{N}(0, I)$. This is because $\mathbb{E}[\epsilon'] = S\Sigma_W^{-1/2} Q \mathbb{E}[\epsilon] = 0$ and

$$
\text{Cov}[\epsilon'] = \mathbb{E}[\epsilon' \epsilon'^T] = S\Sigma_W^{-1/2} Q \mathbb{E}[\epsilon \epsilon^T] Q^T \Sigma_W^{-1/2} S^T = I
$$

As $b = S\Sigma_W^{-1/2} \mu_W$, we can write

$$
\|v\|_F^2 = (\epsilon' + b)^T \Lambda (\epsilon' + b) = \sum_{i=1}^p \eta_i (\epsilon_i' + b_i)^2
$$

Hence each $\epsilon_i' + b_i$ is independently from $\mathcal{N}(b_i, 1)$, and $(\epsilon_i' + b_i)^2$ is independently from the non-central chi-squared distribution of noncentrality parameter $b_i^2$ and with degree 1 of freedom. Thus the MGF of $(\epsilon_i' + b_i)^2$ is

$$
M_{(\epsilon_i' + b_i)^2}(t) = \mathbb{E}_{(\epsilon_i' + b_i)^2}[e^{t(\epsilon_i' + b_i)^2}] = \frac{\exp\left(\frac{b_i^2 t}{1 - 2t}\right)}{(1 - 2t)^{1/2}}
\tag{18}
$$

Let $v_j = y_j - Wx_j$ such that $v_1, v_2, ..., v_m$ are i.i.d. from $\mathcal{N}(\mu_W, \Sigma_W)$, then

$$
R^{\text{emp}}(W) = \frac{1}{m} \sum_{j=1}^m \|y_j - Wx_j\|_F^2 = \frac{1}{m} \sum_{j=1}^m \|v_j\|_F^2
$$

Hence the MGF of $R^{\text{emp}}(W)$ is

$$
\begin{aligned}
M_{R^{\text{emp}}(W)}(t) &= \mathbb{E}_{S \sim \mathcal{D}^m} \left[ e^{t R^{\text{emp}}(W)} \right] = \mathbb{E}_{S \sim \mathcal{D}^m} \left[ \exp\left( \frac{t}{m} \sum_{j=1}^m \|v_j\|_F^2 \right) \right] \\
&= \left( \mathbb{E}_{S \sim \mathcal{D}^m} \left[ \exp\left( \frac{t}{m} \|v\|_F^2 \right) \right] \right)^m = \left( \mathbb{E}_{S \sim \mathcal{D}^m} \left[ \exp\left( \frac{t}{m} \sum_{i=1}^p \eta_i (\epsilon_i' + b_i)^2 \right) \right] \right)^m \\
&= \left( \prod_{i=1}^p \mathbb{E}_{(\epsilon_i' + b_i)^2} \left[ \exp\left( \frac{t\eta_i}{m} (\epsilon_i' + b_i)^2 \right) \right] \right)^m = \left( \prod_{i=1}^p \frac{\exp\left(\frac{tb_i^2 \eta_i}{m - 2t\eta_i}\right)}{(1 - 2t\eta_i/m)^{1/2}} \right)^m \\
&= \frac{\exp\left( \sum_{i=1}^p \frac{tmb_i^2 \eta_i}{m - 2t\eta_i} \right)}{\prod_{i=1}^p (1 - 2t\eta_i/m)^{m/2}}
\end{aligned}
\tag{19}
$$

By Eq (17) and Eq (19), we can expand $\Psi_{\pi, \mathcal{D}}(\lambda, m)$ as

$$\Psi_{\pi, \mathcal{D}}(\lambda, m) = \ln \mathbb{E}_{W \sim \pi} \mathbb{E}_{S \sim \mathcal{D}^m}[e^{\lambda(R^{\text{true}}(W) - R^{\text{emp}}(W))}]$$

$$= \ln \mathbb{E}_{W \sim \pi} \left[ e^{\lambda R^{\text{true}}(W)} \mathbb{E}_{S \sim \mathcal{D}^m}[e^{-\lambda R^{\text{emp}}(W)}] \right]$$

$$= \ln \mathbb{E}_{W \sim \pi} \left[ \exp\left(\lambda \left(\text{tr}(\Sigma_W) + \mu_W^T \mu_W\right)\right) \frac{\exp\left(\sum_{i=1}^p \frac{-\lambda m b_i^2 \eta_i}{m + 2\lambda\eta_i}\right)}{\prod_{i=1}^p (1 + 2\lambda\eta_i/m)^{m/2}} \right] \qquad (20)$$

Use the inequality that for any $x > 0$ and $k > 0$, $e^{\frac{xk}{x+k}} < (\frac{x}{k} + 1)^k$ [1], and the fact $\text{tr}(\Sigma_W) = \sum_{i=1}^p \eta_i$, we have

$$\ln \mathbb{E}_{W \sim \pi} \left[ \exp\left(\lambda \left(\text{tr}(\Sigma_W) + \mu_W^T \mu_W\right)\right) \frac{\exp\left(\sum_{i=1}^p \frac{-\lambda m b_i^2 \eta_i}{m + 2\lambda\eta_i}\right)}{\prod_{i=1}^p (1 + 2\lambda\eta_i/m)^{m/2}} \right]$$

$$\leq \ln \mathbb{E}_{W \sim \pi} \left[ \exp\left(\lambda \left(\text{tr}(\Sigma_W) + \mu_W^T \mu_W\right)\right) \frac{\exp\left(\sum_{i=1}^p \frac{-\lambda m b_i^2 \eta_i}{m + 2\lambda\eta_i}\right)}{\prod_{i=1}^p \exp\left(\frac{m\lambda\eta_i}{m + 2\lambda\eta_i}\right)} \right]$$

$$= \ln \mathbb{E}_{W \sim \pi} \exp\left( \lambda\mu_W^T \mu_W + \sum_{i=1}^p \lambda(\eta_i - \frac{m b_i^2 \eta_i}{m + 2\lambda\eta_i}) - \sum_{i=1}^p \frac{m\lambda\eta_i}{m + 2\lambda\eta_i} \right)$$

$$= \ln \mathbb{E}_{W \sim \pi} \exp\left( \lambda\mu_W^T \mu_W + \sum_{i=1}^p \frac{2\lambda^2\eta_i^2 - \lambda m b_i^2 \eta_i}{m + 2\lambda\eta_i} \right)$$

$$\leq \ln \mathbb{E}_{W \sim \pi} \exp\left( \lambda(\mu_W^T \mu_W - \sum_{i=1}^p b_i^2 \eta_i) + \frac{2\lambda^2(\sum_{i=1}^p \eta_i^2)}{m} \right) = \ln \mathbb{E}_{W \sim \pi} \exp\left( \frac{2\lambda^2(\sum_{i=1}^p \eta_i^2)}{m} \right)$$

The last equality above is because

$$\sum_{i=1}^p b_i^2 \eta_i = b^T \Lambda b = \mu_W^T \Sigma_W^{-1/2} S^T \Lambda S \Sigma_W^{-1/2} \mu_W = \mu_W^T \mu_W$$

Since

$$\sum_{i=1}^p \eta_i^2 = \text{tr}(S^T \Lambda^2 S) = \text{tr}(\Sigma_W^2) = \text{tr}(\Sigma_W \Sigma_W^T) = \|\Sigma_W\|_F^2$$

we have

$$\ln \mathbb{E}_{W \sim \pi} \exp\left( \frac{2\lambda^2(\sum_{i=1}^p \eta_i^2)}{m} \right) = \ln \mathbb{E}_{W \sim \pi} \exp\left( \frac{2\lambda^2 \|\Sigma_W\|_F^2}{m} \right)$$

$\square$

*Proof of Theorem 3.3*:

By Eq (20), we let $\{f_m\}_{m \in \mathbb{N}}$ be a sequence of functions where

$$f_m(W) = \exp\left(\lambda \left(\text{tr}(\Sigma_W) + \mu_W^T \mu_W\right)\right) \frac{\exp\left(\sum_{i=1}^p \frac{-\lambda m b_i^2 \eta_i}{m + 2\lambda\eta_i}\right)}{\prod_{i=1}^p (1 + 2\lambda\eta_i/m)^{m/2}}$$

for $m > 0$, and

$$f_0(W) = \exp\left(\lambda \left(\text{tr}(\Sigma_W) + \mu_W^T \mu_W\right)\right)$$

Note that each $f_i$ is a non-negative function.

---

[1]Since $\frac{x}{x+1} < \ln(x + 1)$ for any $x > -1$, replacing $x$ with $\frac{x}{k}$, and taking exponential on both sides, we get $e^{\frac{xk}{x+k}} < (\frac{x}{k} + 1)^k$.

Now we prove the following three conditions:

(1) $f_m(W) \leq f_0(W)$ for any $m$ and $W$.

Since $\lambda > 0$ and $\eta_i > 0$ for all $i$, we have $f_0(W) \geq f_1(W) \geq f_2(W)...$ for any $W$. This is because, when $W$ is fixed, the numerator $\exp\left(\sum_{i=1}^{p} \frac{-\lambda m b_i^2 \eta_i}{m+2\lambda\eta_i}\right)$ is monotonically decreasing with $m$ for $m \geq 0$, the denominator $\prod_{i=1}^{p}(1+2\lambda\eta_i/m)^{m/2}$ is monotonically increasing with $m$ for $m > 0$, and $(1+2\lambda\eta_i/m)^{m/2} \geq 1$ for any $m > 0$.

(2) $f_m \to 1$ pointwisely as $m \to \infty$.

For any $W$,

$$
\lim_{m\to\infty} f_m(W) = \exp\left(\lambda\left(\text{tr}(\Sigma_W) + \mu_W^T \mu_W\right)\right) \lim_{m\to\infty} \frac{\exp\left(\sum_{i=1}^{p} \frac{-\lambda m b_i^2 \eta_i}{m+2\lambda\eta_i}\right)}{\prod_{i=1}^{p}(1+2\lambda\eta_i/m)^{m/2}}
$$

$$
= \exp\left(\lambda\left(\text{tr}(\Sigma_W) + \mu_W^T \mu_W\right)\right) \frac{\exp\left(\sum_{i=1}^{p} \lim_{m\to\infty} \frac{-\lambda m b_i^2 \eta_i}{m+2\lambda\eta_i}\right)}{\prod_{i=1}^{p} \lim_{m\to\infty}(1+2\lambda\eta_i/m)^{m/2}}
$$

$$
= \exp\left(\lambda\left(\text{tr}(\Sigma_W) + \mu_W^T \mu_W\right)\right) \frac{\exp\left(\sum_{i=1}^{p} -\lambda b_i^2 \eta_i\right)}{\prod_{i=1}^{p} \exp\left(\lambda\eta_i\right)} = 1
$$

The last inequality uses the facts that $\sum_{i=1}^{p} b_i^2 \eta_i = \mu_W^T \mu_W$ and $\sum_{i=1}^{p} \eta_i = \text{tr}(\Sigma_W)$.

(3) $\mathbb{E}[f_0] < \infty$.

$$
\mathbb{E}[f_0] = \mathbb{E}\exp\left(\lambda\left(\text{tr}(\Sigma_W) + \mu_W^T \mu_W\right)\right)
$$

$$
= \mathbb{E}\exp\left(\lambda\left[\text{tr}((W^* - W)\Sigma_x(W^* - W)^T + \Sigma_e) + \|(W^* - W)\mu_x\|_F^2\right]\right)
$$

$$
= \mathbb{E}\exp\left(\lambda\left[\sum_{i=1}^{p}(W^* - W)_{i*}\Sigma_x(W^* - W)_{i*}^T + \text{tr}(\Sigma_e) + \sum_{i=1}^{p}(W^* - W)_{i*}\mu_x\mu_x^T(W^* - W)_{i*}^T\right]\right)
$$

$$
= \mathbb{E}\exp\left(\lambda\left[\sum_{i=1}^{p}(W^* - W)_{i*}\left[\Sigma_x + \mu_x\mu_x^T\right](W^* - W)_{i*}^T + \text{tr}(\Sigma_e)\right]\right)
$$

$$
= \mathbb{E}\exp\left(\lambda\left[\left\|\left(\Sigma_x + \mu_x\mu_x^T\right)^{1/2}(W^* - W)\right\|_F^2 + \text{tr}(\Sigma_e)\right]\right)
$$

$$
= \exp\left(\lambda\text{tr}(\Sigma_e)\right)\mathbb{E}\exp\left(\lambda\left[\left\|\left(\Sigma_x + \mu_x\mu_x^T\right)^{1/2}(W^* - W)\right\|_F^2\right]\right) < \infty
$$

The last inequality holds because $\mathbb{E}\exp\left(\lambda\left[\left\|\left(\Sigma_x + \mu_x\mu_x^T\right)^{1/2}(W^* - W)\right\|_F^2\right]\right) < \infty$ is our assumption and $\exp\left(\lambda\text{tr}(\Sigma_e)\right)$ is a constant.

Denote $E = \mathbb{R}^{p\times p}$ such that $W \in E$. Since $W \sim \pi$, we consider $\pi$ as a probability measure $\mu$ on $E$ with $\mu(E) = 1$. Then we can express $\mathbb{E}[f_m]$ as a Lebesgue integral:

$$
\mathbb{E}[f_m] = \int_E f_m \, d\mu
$$

Also, condition (3) can be written as $\int_E f_0 d\mu < \infty$. Since the conditions (1), (2) and (3) hold, by the Dominated Convergence Theorem (Theorem 11.32, (Rudin, 1976)), we have

$$
\lim_{m\to\infty} \int_E f_m \, d\mu = \int_E \lim_{m\to\infty} f_m \, d\mu = \int_E 1 \, d\mu = 1
$$

Or equivalently,

$$
\lim_{m\to\infty} \mathbb{E}[f_m] = \mathbb{E}\left[\lim_{m\to\infty} f_m\right] = \mathbb{E}[1] = 1
$$

Since $\ln$ is continuous on $(0, \infty)$, we can interchange $\lim$ and $\ln$. Therefore,

$$\lim_{m \to \infty} \Psi_{\pi, \mathcal{D}}(\lambda, m) \leq \lim_{m \to \infty} \ln \mathbb{E}[f_m] = \ln \lim_{m \to \infty} \mathbb{E}[f_m] = \ln 1 = 0$$

$\square$

*Proof of Lemma 3.4*:

Let $X \sim \mathcal{N}(\mu, \sigma^2)$, then for any $t > 0$,

$$\mathbb{E}_X[tY_k] = \int \exp\left(t \sum_{i=0}^{k} a_i x^i\right) \frac{1}{\sqrt{2\pi}\sigma} \exp\left(-\frac{(x-\mu)^2}{2\sigma^2}\right) dx$$

$$= \frac{1}{\sqrt{2\pi}\sigma} \int \exp\left(t \sum_{i=0}^{k} a_i x^i - \frac{(x-\mu)^2}{2\sigma^2}\right) dx \tag{21}$$

Since $k \geq 3$ and $a_k > 0$, $t \sum_{i=0}^{k} a_i x^i - \frac{(x-\mu)^2}{2\sigma^2}$ is a polynomial of $x$ with degree $\geq 3$, with leading coefficient being positive, thus

$$\lim_{x \to \infty} \exp\left(t \sum_{i=0}^{k} a_i x^i - \frac{(x-\mu)^2}{2\sigma^2}\right) = \infty$$

And the integral in Eq (21) is infinity.

$\square$

*Proof of Lemma 4.2*:

$$R^{\text{true}}(W) = \mathbb{E}\left[||y - Wx||_F^2\right] = \sum_{i=1}^{n} \mathbb{E}[||y_i - W_{i*}x||_F^2] = \sum_{i=1}^{n} W_{i*}\mathbb{E}[xx^T]W_{i*}^T - 2W_{i*}\mathbb{E}[y_i x] + \mathbb{E}[y_i^2]$$

$$= \sum_{i=1}^{n} W_{i*}\Sigma_{xx}W_{i*}^T - 2W_{i*}(\Sigma_{xy})_{*i} + (\Sigma_{yy})_{ii}$$

$$= \sum_{i=1}^{n} (W_{i*}\Sigma_{xx}^{1/2})(W_{i*}\Sigma_{xx}^{1/2})^T - 2(W_{i*}\Sigma_{xx}^{1/2})\Sigma_{xx}^{-1/2}(\Sigma_{xy})_{*i} + (\Sigma_{yy})_{ii}$$

$$= \sum_{i=1}^{n} (W_{i*}\Sigma_{xx}^{1/2} - (\Sigma_{xy})_{*i}^T\Sigma_{xx}^{-1/2})(W_{i*}\Sigma_{xx}^{1/2} - (\Sigma_{xy})_{*i}^T\Sigma_{xx}^{-1/2})^T - (\Sigma_{xy})_{*i}^T\Sigma_{xx}^{-1}(\Sigma_{xy})_{*i} + (\Sigma_{yy})_{ii}$$

$$= \sum_{i=1}^{n} ||W_{i*}\Sigma_{xx}^{1/2} - (\Sigma_{xy})_{*i}^T\Sigma_{xx}^{-1/2}||_F^2 - ||\Sigma_{xx}^{-1/2}(\Sigma_{xy})_{*i}||_F^2 + (\Sigma_{yy})_{ii}$$

$$= ||W\Sigma_{xx}^{1/2} - \Sigma_{xy}^T\Sigma_{xx}^{-1/2}||_F^2 - ||\Sigma_{xy}^T\Sigma_{xx}^{-1/2}||_F^2 + \text{tr}(\Sigma_{yy})$$

Since we assume $\Sigma_{xx}$ is positive definite, $\Sigma_{xx}^{-1/2}$ exists.

$\square$

*Proof of Lemma 4.3*:

Since $x = \Delta \odot r$ and $y = (\mathbf{1} - \Delta) \odot r$, we have

$$\Sigma_{xx} = \mathbb{E}\left[xx^T\right] = \mathbb{E}\left[(\Delta \odot r)(\Delta \odot r)^T\right]$$
$$\Sigma_{xy} = \mathbb{E}\left[xy^T\right] = \mathbb{E}\left[(\Delta \odot r)((\mathbf{1} - \Delta) \odot r)^T\right]$$
$$\Sigma_{yy} = \mathbb{E}\left[yy^T\right] = \mathbb{E}\left[((\mathbf{1} - \Delta) \odot r)((\mathbf{1} - \Delta) \odot r)^T\right]$$

We first prove $\Sigma_{xx}$. For $i, j \in \{1, 2, ..., n\}$ and $i \neq j$, $(\Sigma_{xx})_{ij} = \mathbb{E}[\Delta_i \Delta_j r_i r_j]$. Since $\Delta_i \Delta_j r_i r_j$ is a Bernoulli random variable (its value can either be 0 or 1), $\Delta_i$ depends on $r_i$, $\Delta_j$ depends on $r_j$, we have

$$
\begin{aligned}
\mathbb{E}[\Delta_i \Delta_j r_i r_j] &= P\left(\Delta_i \Delta_j r_i r_j = 1\right) = P\left(\Delta_i = 1, \Delta_j = 1, r_i = 1, r_j = 1\right) \\
&= P\left(\Delta_i = 1 | \Delta_j = 1, r_i = 1, r_j = 1\right) P\left(\Delta_j = 1 | r_i = 1, r_j = 1\right) P\left(r_i = 1, r_j = 1\right) \\
&= P\left(\Delta_i = 1 | r_i = 1\right) P\left(\Delta_j = 1 | r_j = 1\right) P\left(r_i = 1, r_j = 1\right) \\
&= p^2 \mathbb{E}[r_i r_j] = p^2 (\Sigma_{rr})_{ij}
\end{aligned}
\tag{22}
$$

For any $i$, $(\Sigma_{xx})_{ii} = \mathbb{E}[(\Delta_i r_i)^2]$. Using the property that a Bernoulli random variable $X$ has $\mathbb{E}[X^2] = \mathbb{E}[X]$,

$$
\begin{aligned}
\mathbb{E}[(\Delta_i r_i)^2] &= P\left(\Delta_i r_i = 1\right) = P\left(\Delta_i = 1, r_i = 1\right) = P\left(\Delta_i = 1 | r_i = 1\right) P\left(r_i = 1\right) = p \mathbb{E}[r_i] \\
&= p \mathbb{E}[r_i^2] = p(\Sigma_{rr})_{ii}
\end{aligned}
\tag{23}
$$

Combining Eq (22) and Eq (23), we get

$$
\Sigma_{xx} = p^2 \Sigma_{rr} + p(1-p)(I \odot \Sigma_{rr})
\tag{24}
$$

Since $(\Sigma_{yy})_{ij} = \mathbb{E}[(1 - \Delta_i)(1 - \Delta_j) r_i r_j]$ and $(\Sigma_{yy})_{ii} = \mathbb{E}[((1 - \Delta_i) r_i)^2]$, replacing $p$ with $1 - p$ in Eq (24), we get $\Sigma_{yy} = (1-p)^2 \Sigma_{rr} + p(1-p)(I \odot \Sigma_{rr})$.

Since $(\Sigma_{xy})_{ij} = \mathbb{E}[\Delta_i(1 - \Delta_j) r_i r_j] = p(1-p)\Sigma_{rr}$ and $(\Sigma_{xy})_{ii} = \mathbb{E}[\Delta_i(1 - \Delta_i) r_i^2] = 0$ (Note that $\Delta_i(1 - \Delta_i) r_i^2 = 0$ regardless of whether $\Delta_i$ is 0 or 1.), we have $\Sigma_{xy} = p(1-p)(\Sigma_{rr} - I \odot \Sigma_{rr})$.

Note that in Eq (24), $I \odot \Sigma_{rr}$ is positive semi-definite and $p^2, p(1-p)$ are positive, thus $\Sigma_{xx}$ is positive definite if $\Sigma_{rr}$ is positive definite.

$\square$

*Proof of Theorem 5.2*:

(1) It is easy to verify that $\mathbb{E}_{W \sim \rho}[W] = \mathcal{U}$ and $\mathbb{E}_{W \sim \rho}[W^T W] = \mathcal{U}^T \mathcal{U} + \text{Diag}\left(\sum_{k=1}^n \mathcal{S}_{k1}, \sum_{k=1}^n \mathcal{S}_{k2}, ..., \sum_{k=1}^n \mathcal{S}_{kn}\right)$. Thus

$$
\mathbb{E}_{W \sim \rho}[R^{\text{emp}}(W)] = \frac{1}{m} \mathbb{E}_{W \sim \rho}[\|Y - WX\|_F^2] = \frac{1}{m} \sum_{l=1}^m \mathbb{E}_{W \sim \rho}[\|Y_{*l} - WX_{*l}\|_F^2]
$$

$$
= \frac{1}{m} \sum_{l=1}^m \mathbb{E}_{W \sim \rho}[(Y_{*l}^T - X_{*l}^T W^T)(Y_{*l} - WX_{*l})] = \frac{1}{m} \sum_{l=1}^m Y_{*l}^T Y_{*l} - 2Y_{*l}^T \mathbb{E}_{W \sim \rho}[W] X_{*l} + X_{*l}^T \mathbb{E}_{W \sim \rho}[W^T W] X_{*l}
$$

$$
= \frac{1}{m} \sum_{l=1}^m Y_{*l}^T Y_{*l} - 2Y_{*l}^T \mathcal{U} X_{*l} + X_{*l}^T \mathcal{U}^T \mathcal{U} X_{*l} + X_{*l}^T \text{Diag}\left(\sum_{k=1}^n \mathcal{S}_{k1}, \sum_{k=1}^n \mathcal{S}_{k2}, ..., \sum_{k=1}^n \mathcal{S}_{kn}\right) X_{*l}
\tag{25}
$$

$D(\rho \| \pi)$ can also be written as a function of $\mathcal{U}$ and $\mathcal{S}$ by

$$
D(\rho \| \pi) = \frac{1}{2}\left[n^2(2 \ln \sigma - 1) - \sum_{k=1}^n \sum_{l=1}^n (\ln \mathcal{S}_{kl} - \frac{\mathcal{S}_{kl}}{\sigma^2}) + \frac{\|\mathcal{U} - \mathcal{U}_0\|_F^2}{\sigma^2}\right]
\tag{26}
$$

Denote $f(\mathcal{U}, \mathcal{S} | \mathcal{U}_0, \sigma, \lambda) = \mathbb{E}_{W \sim \rho}[R^{\text{emp}}(W)] + \frac{1}{\lambda} D(\rho \| \pi)$, our optimization problem becomes

$$
\min_{\mathcal{U}, \mathcal{S}} f(\mathcal{U}, \mathcal{S} | \mathcal{U}_0, \sigma, \lambda)
\tag{27}
$$

The optimal $\mathcal{U}$ and $\mathcal{S}$ has closed-form solution, which can be obtained by solving $\frac{\partial}{\partial \mathcal{U}} f(\mathcal{U}, \mathcal{S} | \mathcal{U}_0, \sigma, \lambda) = 0$ and $\frac{\partial}{\partial \mathcal{S}} f(\mathcal{U}, \mathcal{S} | \mathcal{U}_0, \sigma, \lambda) = 0$.

First we show the partial derivatives of the $\frac{1}{\lambda} D(\rho \| \pi)$ term:

$$
\frac{\partial}{\partial \mathcal{U}_{ij}} \frac{1}{\lambda} D(\rho \| \pi) = \frac{(\mathcal{U}_{ij} - (\mathcal{U}_0)_{ij})}{\lambda \sigma^2}, \quad \frac{\partial}{\partial \mathcal{S}_{ij}} \frac{1}{\lambda} D(\rho \| \pi) = -\frac{1}{2\lambda}\left(\frac{1}{\mathcal{S}_{ij}} - \frac{1}{\sigma^2}\right)
$$

Then we discuss the partial derivatives of the $\mathbb{E}_{W \sim \rho}[R^{\text{emp}}(W)]$ term. By Eq (25), for any $i, j$,

$$\frac{\partial}{\partial \mathcal{S}_{ij}} \mathbb{E}_{W \sim \rho}[R^{\text{emp}}(W)] = \frac{\partial}{\partial \mathcal{S}_{ij}} \frac{1}{m} \sum_{l=1}^{m} X_{*l}^T \text{Diag}\left(\sum_{k=1}^{n} \mathcal{S}_{k1}, \sum_{k=1}^{n} \mathcal{S}_{k2}, ..., \sum_{k=1}^{n} \mathcal{S}_{kn}\right) X_{*l}$$

$$= \frac{\partial}{\partial \mathcal{S}_{ij}} \frac{1}{m} \sum_{l=1}^{m} X_{jl} \mathcal{S}_{ij} X_{jl} = \frac{1}{m} \sum_{l=1}^{m} X_{jl}^2 = \frac{1}{m} X_{j*} X_{j*}^T$$

$$\frac{\partial}{\partial \mathcal{U}_{ij}} \mathbb{E}_{W \sim \rho}[R^{\text{emp}}(W)] = \frac{\partial}{\partial \mathcal{U}_{ij}} \frac{1}{m} \sum_{l=1}^{m} -2Y_{*l}^T \mathcal{U} X_{*l} + X_{*l}^T \mathcal{U}^T \mathcal{U} X_{*l} = \frac{1}{m} \sum_{l=1}^{m} \left(-2Y_{il} X_{jl} + \frac{\partial}{\partial \mathcal{U}_{ij}} \sum_{k=1}^{n} (\mathcal{U}_{k*} X_{*l})^2\right)$$

$$= \frac{1}{m} \sum_{l=1}^{m} \left(-2Y_{il} X_{jl} + \frac{\partial}{\partial \mathcal{U}_{ij}} (\mathcal{U}_{i*} X_{*l})^2\right) = \frac{1}{m} \sum_{l=1}^{m} (-2Y_{il} X_{jl} + 2(\mathcal{U}_{i*} X_{*l}) X_{jl})$$

$$= \frac{2}{m} \left(-Y_{i*} X_{j*}^T + \mathcal{U}_{i*} X X_{j*}^T\right)$$

Wrap up the above results, we get

$$\frac{\partial}{\partial \mathcal{S}_{ij}} f(\mathcal{U}, \mathcal{S} | \mathcal{U}_0, \sigma, \lambda) = \frac{1}{m} X_{j*} X_{j*}^T - \frac{1}{2\lambda}\left(\frac{1}{\mathcal{S}_{ij}} - \frac{1}{\sigma^2}\right) \tag{28}$$

$$\frac{\partial}{\partial \mathcal{U}_{ij}} f(\mathcal{U}, \mathcal{S} | \mathcal{U}_0, \sigma, \lambda) = \frac{2}{m} \left(-Y_{i*} X_{j*}^T + \mathcal{U}_{i*} X X_{j*}^T\right) + \frac{(\mathcal{U}_{ij} - (\mathcal{U}_0)_{ij})}{\lambda \sigma^2} \tag{29}$$

Therefore, the solution of $\frac{\partial}{\partial \mathcal{S}} f(\mathcal{U}, \mathcal{S} | \mathcal{U}_0, \sigma, \lambda) = 0$ is that, for any $i = 1, 2, ..., n$,

$$\mathcal{S}_{ij} = \frac{1}{\frac{2\lambda}{m} X_{j*} X_{j*}^T + \frac{1}{\sigma^2}} \quad \text{for } j = 1, 2, ..., n \tag{30}$$

By Eq (29) we have

$$\frac{\partial}{\partial \mathcal{U}} f(\mathcal{U}, \mathcal{S} | \mathcal{U}_0, \sigma, \lambda) = \left[\frac{2}{m}(-YX^T + \mathcal{U} X X^T) + \frac{1}{\lambda \sigma^2}(\mathcal{U} - \mathcal{U}^0)\right]^T \tag{31}$$

Thus the solution of $\frac{\partial}{\partial \mathcal{U}} f(\mathcal{U}, \mathcal{S} | \mathcal{U}_0, \sigma, \lambda) = 0$ is

$$\mathcal{U} = \left(\frac{1}{m} YX^T + \frac{1}{2\lambda \sigma^2} \mathcal{U}_0\right) \left(\frac{1}{m} X X^T + \frac{1}{2\lambda \sigma^2} I\right)^{-1} \tag{32}$$

Now we show that $f(\mathcal{U}, \mathcal{S} | \mathcal{U}_0, \sigma, \lambda)$ is a convex function, such that the solutions of $\mathcal{S}$ in Eq (30) and $\mathcal{U}$ in Eq (32) are the global minimizer of Eq (27). By Eq (28) and Eq (29) we have

$$\frac{\partial^2 f}{\partial \mathcal{S}_{ij} \partial \mathcal{S}_{kl}} = \begin{cases} \frac{1}{2\lambda (\mathcal{S}_{ij})^2} & \text{if } i = k, j = l \\ 0 & \text{otherwise} \end{cases}, \qquad \frac{\partial^2 f}{\partial \mathcal{U}_{ij} \partial \mathcal{U}_{kl}} = \begin{cases} \frac{2}{m} X_{j*} X_{l*}^T + \frac{1}{\lambda \sigma^2} & \text{if } i = k, j = l \\ \frac{2}{m} X_{j*} X_{l*}^T & \text{if } i = k, j \neq l \\ 0 & \text{otherwise} \end{cases}$$

Denote $\nu \in \mathbb{R}^{2n^2}$ where for $i = 1, 2, ..., n$ and $j = 1, 2, ..., n$, $\nu_{(i-1)n+j} = \mathcal{U}_{ij}$ and $\nu_{n^2+(i-1)n+j} = \mathcal{S}_{ij}$. Let $H_f \in \mathbb{R}^{2n^2 \times 2n^2}$ be the Hessian matrix where $(H_f)_{ij} = \frac{\partial^2 f}{\partial \nu_i \partial \nu_j}$. Then we can write $H_f = \begin{bmatrix} A & 0 \\ 0 & B \end{bmatrix}$ where $A = \frac{2}{m}(XX^T) \otimes I_n + \frac{1}{\lambda \sigma^2} I_{n^2}$ and $B$ is a $n^2 \times n^2$ diagonal matrix with $B_{(i-1)n+j, (i-1)n+j} = \frac{1}{2\lambda (\mathcal{S}_{ij})^2}$. Here $\otimes$ means Kronecker product.

The Kronecker product has a property that, let $\{\lambda_i | i = 1, ..., m\}$ be the eigenvalues of $A \in \mathbb{R}^{m \times m}$ and $\{\mu_j | j = 1, ..., n\}$ be the eigenvalues of $B \in \mathbb{R}^{n \times n}$, then $\{\lambda_i \mu_j | i = 1, ..., m, j = 1, ..., n\}$ are the eigenvalues of $A \otimes B$ (Theorem 4.2.12, (Horn & Johnson, 1991)). Since $XX^T$ is positive semi-definite and $I_n$ is positive definite, $(XX^T) \otimes I_n$ is positive semi-definite.

Thus $A$ is positive definite. Since all elements of $\mathcal{S}$ is positive, $B$ is positive definite. Therefore, $H_f$ is a positive definite matrix for any $\mathcal{U}$ and $\mathcal{S}$, which means $f(\mathcal{U}, \mathcal{S}|\mathcal{U}_0, \sigma, \lambda)$ is a convex function. Thus, the solutions of $\mathcal{S}$ in Eq (30) and $\mathcal{U}$ in Eq (32) give the global minimum.

(2) Applying the constraint diag$(W) = 0$ to $\rho$ and $\pi$ is equivalent to set diag$(\mathcal{U}) = 0$, diag$(\mathcal{S}) = 0$, diag$(\mathcal{U}_0) = 0$, and diag$(\sigma^2 J) = 0$. Under these constraints, the expression of $D(\rho \,||\, \pi)$ in Eq (26) is changed to

$$D(\rho \,||\, \pi) = \frac{1}{2}\left[(n^2 - n)(2\ln\sigma - 1) - \sum_{k=1}^{n}\sum_{l=1, l\neq k}^{n}(\ln\mathcal{S}_{kl} - \frac{\mathcal{S}_{kl}}{\sigma^2}) + \frac{\|\mathcal{U} - \mathcal{U}_0\|_F^2}{\sigma^2}\right] \tag{33}$$

In this case, Eq (28) holds only for $i \neq j$.

We let $\mathcal{S}_{11}, \mathcal{S}_{22}, ..., \mathcal{S}_{nn}$ be zero constants in $f(\mathcal{U}, \mathcal{S}|\mathcal{U}_0, \sigma, \lambda)$, and consider only the off-diagonal elements of $\mathcal{S}$ to be variables. Then we construct the Lagrangian function as

$$L(\mathcal{U}, \mathcal{S}, x|\mathcal{U}_0, \sigma, \lambda) = f(\mathcal{U}, \mathcal{S}|\mathcal{U}_0, \sigma, \lambda) + x^T \text{diag}(\mathcal{U})$$

for some $x \in \mathbb{R}^n$, and solve

$$\frac{\partial L}{\partial x} = [\text{diag}(\mathcal{U})]^T = 0 \tag{34}$$

$$\frac{\partial L}{\partial \mathcal{U}} = \frac{\partial}{\partial \mathcal{U}}f(\mathcal{U}, \mathcal{S}|\mathcal{U}_0, \sigma, \lambda) + \text{Diag}(x) = 0 \tag{35}$$

$$\frac{\partial L}{\partial \mathcal{S}_{ij}} = \frac{\partial}{\partial \mathcal{S}_{ij}}f(\mathcal{U}, \mathcal{S}|\mathcal{U}_0, \sigma, \lambda) = 0 \quad \text{for } i, j \in \{1, 2, ..., n\}, i \neq j \tag{36}$$

The optimal $\mathcal{S}$ is obtained by solving Eq (36) and set $S_{ii} = 0$ for all $i$. The solution of Eq (36) is Eq (30) with $i \neq j$. The optimal $\mathcal{U}$ is obtained by solving Eq (35) and Eq (34). By Eq (35),

$$\frac{2}{m}(-YX^T + \mathcal{U}XX^T) + \frac{1}{\lambda\sigma^2}(\mathcal{U} - \mathcal{U}^0) + \text{Diag}(x) = 0$$

$$\Longleftrightarrow \mathcal{U} = \left(\frac{1}{m}YX^T + \frac{1}{2\lambda\sigma^2}\mathcal{U}_0 - \frac{1}{2}\text{Diag}(x)\right)\left(\frac{1}{m}XX^T + \frac{1}{2\lambda\sigma^2}I\right)^{-1} \tag{37}$$

Then we solve $x$ to satisfy Eq (34),

$$\text{diag}(\mathcal{U}) = \text{diag}\left[\left(\frac{1}{m}YX^T + \frac{1}{2\lambda\sigma^2}\mathcal{U}_0\right)\left(\frac{1}{m}XX^T + \frac{1}{2\lambda\sigma^2}I\right)^{-1}\right] - \text{diag}\left[\frac{1}{2}\text{Diag}(x)\left(\frac{1}{m}XX^T + \frac{1}{2\lambda\sigma^2}I\right)^{-1}\right]$$

$$= \text{diag}\left[\left(\frac{1}{m}YX^T + \frac{1}{2\lambda\sigma^2}\mathcal{U}_0\right)\left(\frac{1}{m}XX^T + \frac{1}{2\lambda\sigma^2}I\right)^{-1}\right] - \frac{1}{2}x \odot \text{diag}\left[\left(\frac{1}{m}XX^T + \frac{1}{2\lambda\sigma^2}I\right)^{-1}\right] = 0$$

we get

$$x = 2 \cdot \text{diag}\left[\left(\frac{1}{m}YX^T + \frac{1}{2\lambda\sigma^2}\mathcal{U}_0\right)\left(\frac{1}{m}XX^T + \frac{1}{2\lambda\sigma^2}I\right)^{-1}\right] \oslash \text{diag}\left[\left(\frac{1}{m}XX^T + \frac{1}{2\lambda\sigma^2}I\right)^{-1}\right]$$

Now we show that the solution of Eq (34), Eq (35) and Eq (36) gives the global minimum of the problem Eq (27) under the constraint diag$(W) = 0$. Let $H_L$ be the Hessian matrix of $L$. It is easy to verify that if we remove the dimensions corresponding to $\mathcal{S}_{11}, \mathcal{S}_{22}, ...\mathcal{S}_{nn}$ of $H_f$ and get $H_f' \in \mathbb{R}^{(2n^2-n)\times(2n^2-n)}$, then $H_L$ will be equivalent to $H_f'$. Thus $H_L$ is positive definite for any $\mathcal{U}, \mathcal{S}$.

We use the second order sufficiency conditions (Section 11.5, (Luenberger & Ye, 2008)): Let $(\mathcal{U}^*, \mathcal{S}^*, x^*)$ be a solution of $\frac{\partial L}{\partial \mathcal{U}} = 0, \frac{\partial L}{\partial \mathcal{S}} = 0, \frac{\partial L}{\partial x} = 0$, then $(\mathcal{U}^*, \mathcal{S}^*)$ is a local minimizer of $f$ if $H_L|_{\mathcal{U}=\mathcal{U}^*, \mathcal{S}=\mathcal{S}^*, x=x^*}$ is positive semi-definite on the subspace $M = \{y \in \mathbb{R}^{2n^2-n} \,|\, y_{(i-1)n+i} = 0 \text{ for } i = 1, 2, ..., n\}$ (The subspace requires $[(\frac{\partial \mathcal{U}_{11}}{\partial v})^T, (\frac{\partial \mathcal{U}_{22}}{\partial v})^T, ..., (\frac{\partial \mathcal{U}_{nn}}{\partial v})^T]^T y = 0$, i.e., $\frac{\partial \mathcal{U}_{ii}}{\partial v}y = y_{(i-1)n+i} = 0$ for all $i$). We have shown that the solution of Eq (34),

Eq (35) and Eq (36) is unique, and the $H_L$ with respect to this solution is positive definite on the entire space $\mathbb{R}^{2n^2-n}$. Since $M$ is a subspace of $\mathbb{R}^{2n^2-n}$, the positive definiteness of $H_L$ holds on $M$, thus the second order sufficiency condition is satisfied. Therefore, this solution gives the global minimum.

$\square$

*Proof of Proposition 5.3*:

Denote $v_i = (W_{i*}\Sigma_{xx}^{1/2} + B_{i*})^T$, then $v_i = A^{1/2}\epsilon + \mu^i$ where $\epsilon \in \mathcal{N}(0, I)$. Using the quadratic form shown in Theorem 3.2, we have

$$\|v_i\|_F^2 = (A^{1/2}\epsilon + \mu^i)^T(A^{1/2}\epsilon + \mu^i) = (A^{1/2}\epsilon + \mu^i)^T A^{-1/2}S^T\Lambda S A^{-1/2}(A^{1/2}\epsilon + \mu^i)$$

$$= (S\epsilon + SA^{-1/2}\mu^i)^T\Lambda(S\epsilon + SA^{-1/2}\mu^i) = (S\epsilon + \bar{b}^i)^T\Lambda(S\epsilon + \bar{b}^i) = \sum_{j=1}^{n}\eta_j(S_{j*}\epsilon + \bar{b}^i_j)^2$$

It is easy to show that each $S_{j*}\epsilon$ are i.i.d. from $\mathcal{N}(0, 1)$ for all $j$, thus each $S_{j*}\epsilon + \bar{b}^i_j$ is independently from $\mathcal{N}(\bar{b}^i_j, 1)$. Since each $v_i$ is independent, Eq (11) can be rewritten as

$$\mathbb{E}_\pi\left[e^{\lambda R^{\text{true}}(W)}\right] = C\,\mathbb{E}_\pi\left[e^{\lambda\sum_{i=1}^{n}\|W_{i*}\Sigma_{xx}^{1/2}+B_{i*}\|_F^2}\right] = C\prod_{i=1}^{n}\mathbb{E}_\pi\left[e^{\lambda\|v_i\|_F^2}\right] = C\prod_{i=1}^{n}\prod_{j=1}^{n}\mathbb{E}_\pi\left[e^{\lambda\eta_j(S_{j*}\epsilon+\bar{b}^i_j)^2}\right]$$

$$= C\prod_{i=1}^{n}\prod_{j=1}^{n}\frac{\exp\left(\frac{\lambda(\bar{b}^i_j)^2\eta_j}{1-2\lambda\eta_j}\right)}{(1-2\lambda\eta_j)^{1/2}}$$

The last equality above follows from Eq (18).

$\square$

*Proof of Theorem 5.4*:

Let $P, Q \in \mathbb{R}^{n\times n}$ be two symmetric matrices, we write $P \succeq Q$ if $P - Q$ is positive semi-definite and $P \succ Q$ if $P - Q$ is positive definite.

Let $\eta_j$ be the $j$th largest eigenvalue of $A$ and $\eta_j^{(i)}$ be the $j$th largest eigenvalue of $A^{(i)}$. By Corollary 7.7.4 (c) of (Horn & Johnson, 2012), $P \succeq Q$ implies $\eta_j(P) \geq \eta_j(Q)$ for any $j$. Since $A - A^{(i)} = \sigma^2(\Sigma_{xx}^{1/2})_{*i}(\Sigma_{xx}^{1/2})_{*i}^T \succeq 0$ for any $i$, we have $\eta_j \geq \eta_j^{(i)}$ for any $i, j$.

Since $b^{(i)} = S^{(i)}(A^{(i)})^{-1/2}\mu^i$, we have

$$(b_j^{(i)})^2\eta_j^{(i)} = \eta_j^{(i)}(\mu^i)^T(A^{(i)})^{-1/2}(S_{j*}^{(i)})^T S_{j*}^{(i)}(A^{(i)})^{-1/2}\mu^i$$

$$= \eta_j^{(i)}(\mu^i)^T(S^{(i)})^T(\Lambda^{(i)})^{-1/2}[S^{(i)}(S_{j*}^{(i)})^T][S_{j*}^{(i)}(S^{(i)})^T](\Lambda^{(i)})^{-1/2}(S^{(i)})\mu^i$$

$$= (\mu^i)^T(S_{j*}^{(i)})^T(S_{j*}^{(i)})\mu^i$$

Therefore, Eq (13) can be expressed as

$$\frac{1}{C}\mathbb{E}_{\pi'}\left[e^{\lambda R^{\text{true}}(W)}\right] = \prod_{i=1}^{n}\prod_{j=1}^{n}\frac{\exp\left(\frac{\lambda(b_j^{(i)})^2\eta_j^{(i)}}{1-2\lambda\eta_j}\right)}{\left(1-2\lambda\eta_j^{(i)}\right)^{1/2}} = \prod_{i=1}^{n}\prod_{j=1}^{n}\frac{\exp\left(\frac{\lambda(\mu^i)^T(S_{j*}^{(i)})^T(S_{j*}^{(i)})\mu^i}{1-2\lambda\eta_j}\right)}{\left(1-2\lambda\eta_j^{(i)}\right)^{1/2}}$$

$$= \prod_{i=1}^{n}\frac{\exp\left(\lambda(\mu^i)^T\left(\sum_{j=1}^{n}\frac{(S_{j*}^{(i)})^T(S_{j*}^{(i)})}{1-2\lambda\eta_j}\right)\mu^i\right)}{\prod_{j=1}^{n}\left(1-2\lambda\eta_j^{(i)}\right)^{1/2}} = \prod_{i=1}^{n}\frac{\exp\left(\lambda(\mu^i)^T(S^{(i)})^T\bar{\Lambda}^{(i)}S^{(i)}\mu^i\right)}{\prod_{j=1}^{n}\left(1-2\lambda\eta_j^{(i)}\right)^{1/2}}$$

where $\bar{\Lambda}^{(i)} = \text{diag}\left(\frac{1}{1-2\lambda\eta_1^{(i)}}, \frac{1}{1-2\lambda\eta_2^{(i)}}, ..., \frac{1}{1-2\lambda\eta_n^{(i)}}\right)$.

Similarly, Eq (12) can be expressed as

$$\frac{1}{C} \mathbb{E}_\pi \left[ e^{\lambda R^{\text{true}}(W)} \right] = \prod_{i=1}^n \frac{\exp \left( \lambda(\mu^i)^T S^T \bar{\Lambda} S \mu^i \right)}{\prod_{j=1}^n \left( 1 - 2\lambda\eta_j \right)^{1/2}}$$

where $\bar{\Lambda} = \text{diag} \left( \frac{1}{1-2\lambda\eta_1}, \frac{1}{1-2\lambda\eta_2}, ..., \frac{1}{1-2\lambda\eta_n} \right)$.

Now we show that $S^T \bar{\Lambda} S \succeq (S^{(i)})^T \bar{\Lambda}^{(i)} S^{(i)}$ for any $i$. By Corollary 7.7.4 (a) of (Horn & Johnson, 2012), if $P \succ 0$ and $Q \succ 0$, then $P \succeq Q$ if and only if $Q^{-1} \succeq P^{-1}$. Since we assume $0 < \lambda < \frac{1}{2\eta_1}$, we have $1 - 2\lambda\eta_j^{(i)} > 0$ and $1 - 2\lambda\eta_j > 0$ for any $i, j$, thus all diagonal elements of $\bar{\Lambda}^{(i)}$ and $\bar{\Lambda}$ are positive, implying that $(S^{(i)})^T \bar{\Lambda}^{(i)} S^{(i)} \succ 0$ and $S^T \bar{\Lambda} S \succ 0$.

Since $\left( (S^{(i)})^T \bar{\Lambda}^{(i)} S^{(i)} \right)^{-1} = (S^{(i)})^T \left( I - 2\lambda\Lambda^{(i)} \right) S^{(i)} = I - 2\lambda A^{(i)}$ and $\left( S^T \bar{\Lambda} S \right)^{-1} = I - 2\lambda A$, we have

$$\left( (S^{(i)})^T \bar{\Lambda}^{(i)} S^{(i)} \right)^{-1} \succeq \left( S^T \bar{\Lambda} S \right)^{-1} \iff I - 2\lambda A^{(i)} \succeq I - 2\lambda A \iff A \succeq A^{(i)}$$

Thus $S^T \bar{\Lambda} S \succeq (S^{(i)})^T \bar{\Lambda}^{(i)} S^{(i)}$, implying that $(\mu^i)^T S^T \bar{\Lambda} S \mu^i \geq (\mu^i)^T (S^{(i)})^T \bar{\Lambda}^{(i)} S^{(i)} \mu^i$ holds for any $\mu^i$. Therefore,

$$\frac{1}{C} \mathbb{E}_{\pi'} \left[ e^{\lambda R^{\text{true}}(W)} \right] = \prod_{i=1}^n \frac{\exp \left( \lambda(\mu^i)^T (S^{(i)})^T \bar{\Lambda}^{(i)} S^{(i)} \mu^i \right)}{\prod_{j=1}^n \left( 1 - 2\lambda\eta_j^{(i)} \right)^{1/2}} \leq \prod_{i=1}^n \frac{\exp \left( \lambda(\mu^i)^T S^T \bar{\Lambda} S \mu^i \right)}{\prod_{j=1}^n \left( 1 - 2\lambda\eta_j \right)^{1/2}} = \frac{1}{C} \mathbb{E}_\pi \left[ e^{\lambda R^{\text{true}}(W)} \right]$$

$\square$

## B. Allowing Multiple Trails on $\lambda$

Since we do not know the optimal value of $\lambda$, by Section 2.1.4 of (Alquier, 2021), we can choose a finite grid in $(0, +\infty)$ and search $\lambda$ in the grid. Let $\Lambda = \{\lambda_1, \lambda_2, ..., \lambda_L\}$ be the grid where each $\lambda_i > 0$ and $L$ is the cardinality of $\Lambda$.

$$P \left( \forall \lambda \in \Lambda, \quad \mathbb{E}_{W \sim \rho}[R^{\text{true}}(W)] < \mathbb{E}_{W \sim \rho}[R^{\text{emp}}(W)] + \frac{1}{\lambda} \left[ D(\rho \| \pi) + \ln \frac{L}{\delta} + \Psi_{\pi, \mathcal{D}}(\lambda, m) \right] \right) \geq 1 - \delta$$

This is because

$$P \left( \forall \lambda \in \Lambda, \quad \mathbb{E}_{W \sim \rho}[R^{\text{true}}(W)] < \mathbb{E}_{W \sim \rho}[R^{\text{emp}}(W)] + \frac{1}{\lambda} \left[ D(\rho \| \pi) + \ln \frac{L}{\delta} + \Psi_{\pi, \mathcal{D}}(\lambda, m) \right] \right)$$

$$= 1 - P \left( \exists \lambda \in \Lambda, \quad \mathbb{E}_{W \sim \rho}[R^{\text{true}}(W)] > \mathbb{E}_{W \sim \rho}[R^{\text{emp}}(W)] + \frac{1}{\lambda} \left[ D(\rho \| \pi) + \ln \frac{L}{\delta} + \Psi_{\pi, \mathcal{D}}(\lambda, m) \right] \right)$$

$$= 1 - P \left( \bigcup_{i=1}^L \mathbb{E}_{W \sim \rho}[R^{\text{true}}(W)] > \mathbb{E}_{W \sim \rho}[R^{\text{emp}}(W)] + \frac{1}{\lambda_i} \left[ D(\rho \| \pi) + \ln \frac{L}{\delta} + \Psi_{\pi, \mathcal{D}}(\lambda_i, m) \right] \right)$$

$$\geq 1 - \sum_{i=1}^L P \left( \mathbb{E}_{W \sim \rho}[R^{\text{true}}(W)] > \mathbb{E}_{W \sim \rho}[R^{\text{emp}}(W)] + \frac{1}{\lambda_i} \left[ D(\rho \| \pi) + \ln \frac{L}{\delta} + \Psi_{\pi, \mathcal{D}}(\lambda_i, m) \right] \right)$$

$$\geq 1 - \sum_{i=1}^L \frac{\delta}{L} = 1 - \delta$$

## C. Related Works

The earliest PAC-Bayes bound is proposed by (McAllester, 1998). (Alquier et al., 2016) proposed an oracle PAC-Bayes bound based under Hoeffding assumption. (Germain et al., 2016) applied Alquier's bound to linear regression problem under Gaussian data and parameter distribution assumptions, but the bound does not converge for being independent of the number of samples. (Shalaeva et al., 2020) improved Germain's bound by proposing a bound related to the number of samples, and showed the bound converges as the number of samples increases. Most PAC-Bayes bounds are theoretical and difficult to calculate in practice, and some research is focused on making the bound more practical to compute. (Dziugaite

& Roy, 2017) proposed a practical way to calculate Seeger's bound (Langford & Seeger, 2001) for neural networks, and showed the bound is nonvacuous on MNIST dataset, where the bound is around $10\times$ of the test error.

Recent years LAEs gains popularity in recommendation systems (particularly on implicit settings) due to their simplicity and effectiveness. (Steck, 2019) proposed the EASE model and showed it surpasses the performance of deep neural network models on recommendation datasets under Recall and NDCG metrics. Later (Steck, 2020) proposed EDLAE which introduces a mask to the target function to avoid the parameter matrix overfitting towards identity. (Vančura et al., 2022) proposed ELSA which constructs the LAE with an item-item similarity matrix $AA^T - I$ with zero diagonal. An earlier LAE model SLIM (Ning & Karypis, 2011) obtains the weight matrix by solving an $L_1$ norm and $L_2$ norm optimization problem and under zero diagonal constraint.

Most LAE based recommender models constraints the diagonal of the weight matrix to zero. The zero diagonal constraint is closely related to the trace norm, which is considered an effective tool for matrix completion. (Srebro & Salakhutdinov, 2010) applied the weighted traced norm in collaborative filtering. (Shamir & Shalev-Shwartz, 2014) proposed a sample complexity bound for the trace norm in matrix completion.

Another type of linear recommendation model is based on matrix factorization, which can be viewed as a form of low-rank matrix completion (Candes & Tao, 2009; Recht, 2011; Chen et al., 2014; Srebro & Shraibman, 2005; Foygel et al., 2011; Shamir & Shalev-Shwartz, 2011). Matrix factorization methods have been shown to be highly effective in explicit settings (Koren et al., 2009), where user preferences are explicitly expressed (e.g., ratings). However, they have been found to be less effective than LAE models in implicit settings (Cremonesi & Jannach, 2021; Jin et al., 2021), where interactions are inferred from user behavior (e.g., clicks or purchases).

Some studies have investigated the generalizability of the matrix factorization models. (Srebro et al., 2004) proposed a PAC bound based on covering number for collaborative filtering. Other generalization bounds include (Ledent et al., 2021) for inductive matrix completion and (Ledent & Alves, 2024) for deep non-linear matrix completion.

## D. Supplemental Experiment Results

*Table 3.* Details of the terms of each RH in Table 2

| | Models | ML 20M | Netflix | MSD |
|---|---|---|---|---|
| $\gamma = 50$ | $\lambda$ | 512 | 512 | 512 |
| | $\mathbb{E}_{W \sim \rho}[R^{\text{emp}}(W)]$ | 66.99 | 90.87 | 16.58 |
| | $D(\rho \,\|\, \pi)$ | 0.28 | 0.18 | 0.0019 |
| | $\ln \mathbb{E}_\pi \left[ e^{\lambda R^{\text{true}}(W)} \right]$ | 31571.14 | 44659.37 | 8196.30 |
| $\gamma = 100$ | $\lambda$ | 512 | 512 | 512 |
| | $\mathbb{E}_{W \sim \rho}[R^{\text{emp}}(W)]$ | 65.14 | 89.68 | 16.34 |
| | $D(\rho \,\|\, \pi)$ | 0.27 | 0.17 | 0.0018 |
| | $\ln \mathbb{E}_\pi \left[ e^{\lambda R^{\text{true}}(W)} \right]$ | 31102.53 | 44313.39 | 8141.72 |
| $\gamma = 200$ | $\lambda$ | 512 | 512 | 512 |
| | $\mathbb{E}_{W \sim \rho}[R^{\text{emp}}(W)]$ | 63.59 | 88.57 | 16.12 |
| | $D(\rho \,\|\, \pi)$ | 0.26 | 0.17 | 0.0018 |
| | $\ln \mathbb{E}_\pi \left[ e^{\lambda R^{\text{true}}(W)} \right]$ | 30753.19 | 44014.86 | 8092.62 |

## E. Discussions

Our PAC-Bayes bound can be generalized to any LAE model $W$, including those with the zero diagonal constraint. However, one limitation of our bound is that some LAE models may impose specific constraints on $W$, making the bound difficult to compute. For example, some LAE models require $W$ to be of low rank. While the bound can still be formed for a low rank $W$, its calculation can be difficult (We may end up with a theoretical bound that cannot be computed). This is because, the calculation of the bound relies Assumption 5.1 in our paper, where we assume $W$ to be a random Gaussian matrix ($W$ does not have to be Gaussian, but assuming it to be Gaussian makes the bound easy to calculate, while other distributions may not). A random Gaussian matrix $W$ is of full rank (Feng & Zhang, 2007) and cannot be reduced to the low rank case.

Another limitation is that our bound uses squared loss (squared Frobenius norm) as the metric for the error between target and prediction. The square loss is easy for statistical analysis. However, real-world recommender systems typically use other metrics like NDCG@K, Recall@K for evaluation, and our bound cannot be directly applied to these metrics. Moreover, these metrics are challenging to analyze from a statistical perspective.

