# OpenReview forum: "PAC-Bayes Bounds for Multivariate Linear Regression and Linear Autoencoders"
_ICML.cc/2025/Conference — Submitted to ICML 2025_

### Official Review · Reviewer_8qwC · 2025-03-09

**Overall Recommendation:** 3

**Summary:**

Based on the PAC-Bayes risk bound of Alquier et al. (2016), the authors propose a risk bound for multivariate linear regression (thm 3.2). They extend this result (lem 4.2 and thm 5.2) to linear auto encoders which performs L2-regularized multivariate linear regression under the constraint that the weight matrix has zero diagonal elements. Their risk bounds is a generalization of the risk bound of Shalaeva et al. 2020 and is valid only in the case when data-generating distribution D is given by a multivariate gaussian (thm 3.2) or, more generally, when D is given by a set of cross-correlation matrices (assumption 4.1 for lem 4.2). Their thm 5.2 specifies the analytic form of the optimal Gaussian posterior under assumption 5.1. They also propose a method to compute a tight upper-bound on the log of the moment generating function, \psi, when the data-generating distribution is subjected to assumption 4.1 and when the posterior and prior are subjected to assumption 5.1. Their thm 5.4 states that if the posterior forces the random W to have zero diagonal elements, then \psi is upper bounded by the one when this constraint (of zero diagonal element) is not applied. The algorithm for computing their PAC-Bayes bound is provided by Algorithm 1 which has been applied to 3 large data sets used for benchmarking collaborative filtering methods.

**Claims And Evidence:**

Their theoretical claims are proven in detail. So, unless an undetected error occured, the proofs rigorously support the theoretical claims. However, I have not understood exactly how the authors have applied algorithm 1 for the computation of their PAC-Bayes bounds in their experimental section (Section 6). More details should be provided to convince me about the quality of their risk bound in practice. More importantly, the novelty of their risk bound comes from the assumption that the data-generating distribution is known, which is never the case in practice. That said, such oracle bounds can be very instructive at giving us insight on how to obtain better learning guarantees. However, I would use synthetic data generated according to the given data-generating distribution for the purpose of verifying the tightness of an oracle bound.

**Essential References Not Discussed:**

No.

**Experimental Designs Or Analyses:**

As I have already written, I have not understood exactly how the authors have applied algorithm 1 for the computation of their PAC-Bayes bounds in their experimental section (Section 6). More details should be provided to convince me about the quality of their risk bound in practice. In particular, why did you split the data set into a training and a testing set? A risk bound should be computed on the training set only, not the testing set. See Langford, JMLR 2005, "Tutorial on Practical Prediction Theory for Classification". Moreover, it should be computed for the stochastic Gibbs predictor, not for the deterministic predictor W returned by the EASE optimization problem of equation 2.

**Methods And Evaluation Criteria:**

As I have just written, I would use synthetic data generated according to the given data-generating distribution for the purpose of verifying the tightness of an oracle bound.

**Other Comments Or Suggestions:**

No.

**Other Strengths And Weaknesses:**

As I have written, the novelty of their risk bound comes from the assumption that the data-generating distribution is known, which is never the case in practice. That said, such oracle bounds can be very instructive at giving us insight on how to obtain better learning guarantees. However, I would use synthetic data generated according to the given data-generating distribution for the purpose of verifying the tightness of an oracle bound.

After reading the author's rebuttal and realizing an important methodological flaw, I am lowering my score.

-----

After discussing with reviewers, I do see that W in Algorithm 1 is used to set the mean prior, which is perfectly ok. So I raise my score accordingly, but the authors should fix the formulations of their risk bounds (\forall \rho is inside the probability, not outside).

**Questions For Authors:**

See above.

**Relation To Broader Scientific Literature:**

Yes.

**Theoretical Claims:**

All risk bound theorems provided in this paper are *incorrectly* stated! But that can be corrected very easily and it seems that these errors did not have further implications. Indeed, the statement of Alquier's bound at the beginning of Section 2 is stated as a "test-set" bound because it states that for any \pi and any \rho we have that with prob. at least 1-\delta on the random draws of the training set that the true risk is less than the empirical risk + the confidence interval. However Alquier's THM is much *stronger* that this statement! It states that for any \pi we have that with prob. at least 1-\delta on the random draws of the training set that *for all \rho absolutely continuous with respect to \pi : the true risk is less than the empirical risk + the confidence interval. Do you see the difference between the two statements? The last statement says the upper bounds applies simultaneously for all posterior \rho, including the one returned by the learning algorithm. This is not the same for the first statement which applies only to a single \rho whatever the one chosen in advanced, before looking at the data: it does not applies to a \rho chosen by the learner as a function of the training set.

As a consequence, to correct your statement for Alquier's bound you can insert \forall \rho << \pi: inside the probability, i.e., just after P(  , and you must also remove your statement "and \rho be the posterior distribution" before the inequality. The << symbol means "absolutely continuous with respect to" and it is there to assure that the KL divergence is always defined.  For the same reason, you should include \rho << \pi in Equation 3 and Equation 6.

---

> ### Author Rebuttal · Authors · 2025-04-01
>
> **Theoretical Claims**: All risk bound theorems provided in this paper are incorrectly stated... Alquier's THM is much stronger that this statement. It holds for all $\rho$ and does not require $\rho$ be any specific distribution.
>
> **Answer**: We appreciate you pointing out this issue. The original Alquier's bound [1] is indeed of the form $P(\forall \rho, ...) > 1 - \delta$. We followed Shalaeva's setting $\forall \rho, P(...) > 1 - \delta$ but overlooked this issue.
>
> Let $\textsf{E}(\rho)$ be an event related to $\rho$. It is true that $P(\forall \rho, \textsf{E}(\rho)) > 1 - \delta \Longrightarrow \forall \rho, P(\textsf{E}(\rho)) > 1 - \delta$, since the probability on the left applies to an intersection of events while the probability on the right applies to a single event. Thus, Alquier's bound makes a stronger claim; we will revise its statement accordingly. Shalaeva's bound and our bounds remain valid, as they are derived from a weaker form of Alquier's bound; we think fixing their statements is unnecessary.
>
> **Experimental Designs Or Analyses**: A risk bound should be computed on the training set only, not the testing set. See Langford, JMLR 2005, ``Tutorial on Practical Prediction Theory for Classification''.
>
> **Answer**: In Langford's setting, the same loss function is applied to both training set and test set, so both training error and test error are empirical risks. However, in some cases, training and testing metrics differ. For example, recommender system models like EASE or EDLAE are trained by minimizing an averaged loss (e.g., squared Frobenius norm), but evaluated using an averaged ranking (e.g., Recall@K and NDCG@K).
>
> Since both averaged loss and averaged ranking are evaluated on finite samples, they represent two different types of empirical risk. However, **as the ultimate goal is to enhance model performance on the test set, the empirical risk on the test set is typically the most relevant**. In this case, it is sufficient to develop a generalization bound solely for the test set. Some generalization bounds have even been formulated directly on test metrics like AUC [2].
>
> **Experimental Designs Or Analyses**: I have not understood exactly how the authors have applied algorithm 1 for the computation of their PAC-Bayes bounds in their experimental section (Section 6). ... Moreover, it should be computed for the stochastic Gibbs predictor, not for the deterministic predictor W returned by the EASE optimization problem of equation 2.
>
> **Answer**: In our experiment, after obtaining a deterministic predictor $W$ by solving the EASE problem, we set it as the expectation of $\pi$, i.e., $\pi = \bar{\mathcal{N}}(W, \sigma^2J)$, similar to the approach of Dziugaite and Roy [3]. This $\pi$ is then fed into Algorithm 1 to solve for the optimal $\rho$. The predictor in our bound is **always stochastic**, as both its prior distribution $\pi$ and posterior distribution $\rho$ are Gaussian. While a Gaussian $\rho$ may not be optimal (i.e., it may not approximate the Gibbs posterior [4]), we believe it is sufficient to verify the tightness of the bound.
>
> **Other Strengths And Weaknesses**: The novelty of their risk bound comes from the assumption that the data-generating distribution is known, which is never the case in practice.
>
> **Answer**: We use an unbounded loss in our linear regression setting. To our knowledge, it is typical to assume that the data-generating distribution is known when deriving a PAC-Bayes bound for unbounded losses, as seen in two recent works [5][6]. This is because unbounded losses require assumptions (such as a bounded CGF [4]) to characterize their tail distributions, and without knowing the data distribution, such assumptions in general cannot be formed. This nature makes such bounds inevitably oracle-based.
>
> While most of such oracle bounds remain theoretical, our approach shows that they can be computed with certain assumptions applied, and we have identified which assumptions simplify the computation.  Furthermore, we have shown that the computation can be highly efficient, as evidenced by empirical validation on large real-world datasets. Although not fully practical yet, we believe our computational methods can inspire future work in this area.
>
> **References**:
>
> [1] Alquier et al. On the properties of variational approximations of gibbs posteriors. JMLR 2016.
>
> [2] Shivani Agarwal et al. Generalization Bounds for the Area Under the ROC Curve. JMLR, 2005.
>
> [3] Dziugaite and Roy. Computing nonvacuous generalization bounds for deep (stochastic) neural networks with many more parameters than training data. arXiv, 2017.
>
> [4] Borja Rodrıguez-Galvez et al. More PAC-Bayes bounds: From bounded losses, to losses with general tail behaviors, to anytime validity. JMLR, 2024.
>
> [5] Maxime Haddouche et al. PAC-Bayes unleashed: Generalisation bounds with unbounded losses. Entropy, 2021.
>
> [6] Ioar Casado et al. PAC-Bayes-Chernoff bounds for unbounded losses. NIPS, 2024.

---

> > ### Comment · Reviewer_8qwC · 2025-04-04
> >
> > The authors mention that since Alquier's original bound is a stronger form than their formulation written at the beginning of section 2, than no modification of their statement is necessary. I disagree with this since the prior, in that case, does not play any role and this implies the the KL divergence should be absent in their formulation of the bound. This also means that the KL divergence should also be absent in every risk bound written in their paper since, in all their statements the "\forall \rho" is outside of the probability (of the random draws of the training set). Adding a positive quantity like the KL divergence in the risk bound still makes it valid but it also makes it much looser and irrelevant if the empirical risk of the returned posterior (from the training set) is evaluated on the test set! Admittedly in their rebuttal, this is exactly what they have done in their Algorithm 1 to compute the bound. I was not sure about this (while I was reading their paper) as I would have never though that they would make this error. But this is what they did! Note that Shalaeva's et al. (2020) risk bounds have also been incorrectly written but without anymore implications (except of just correcting risk the bound statements) for the correctness of their paper. This is in sharp contrast with the current paper which evaluates a training set bound bound on a test set. Consequently, I lower my evaluation score accordingly.

---

> > > ### Author Response · Authors · 2025-04-05
> > >
> > > Thanks for your clarification. We apologize for the lack of clarity in our previous answer due to a 5000-word limit.
> > >
> > > **On the "methodological flaw"**:
> > >
> > > We believe that our computing method for the PAC-Bayes bound does not contain a "methodological flaw". The reason is as follows:
> > >
> > > Consider four different statements of PAC-Bayes bounds:
> > > \begin{align*}
> > >     &\text{S1}: \\; P(\forall \rho, \textsf{E}(\rho)) > 1 - \delta \\\\
> > >     &\text{S2}: \\; \forall \rho, P(\textsf{E}(\rho)) > 1 - \delta \\\\
> > >     &\text{S3}: \\; \rho \text{ is a Gibbs posterior}, P(\textsf{E}(\rho)) > 1 - \delta \\\\
> > >     &\text{S4}: \\; \rho \text{ is a Guassian posterior}, P(\textsf{E}(\rho)) > 1 - \delta
> > > \end{align*}
> > >
> > > It is easy to see that $\text{S1} \Longrightarrow \text{S2}, \text{S2} \Longrightarrow \text{S3}, \text{S2} \Longrightarrow \text{S4}$. $\text{S1} \Longrightarrow \text{S2}$ is because $P(\forall \rho, \textsf{E}(\rho)) = P(\cap\_\{\rho\} \textsf{E}(\rho)) \le P(\textsf{E}(\rho\_0))$ holds for any single $\rho\_0$, as $\cap\_\{\rho\} \textsf{E}(\rho) \subset \textsf{E}(\rho\_0)$. The bounds in our paper are of the form S2 in Section 2, 3, 4, and of the form S4 in Section 5.
> > >
> > > We understand that optimizing $\rho$ on S3 -- searching for the Gibbs posterior $\rho$ -- is likely what you consider the methodologically correct approach. The Gibbs posterior is known to minimize the right hand side of Alquier's bound, making it optimal. However, it is typically non-computable, and approximating it requires sampling-based methods. Sampling can be computationally inefficient on large datasets. For example, on the MSD dataset used in our experiments, even with a high performance GPU, calculating a single sample takes around 10 minutes.
> > >
> > > To reduce computational costs, one practical alternative is to use S4 instead of S3, as the bound is in general easier to compute with a Gaussian posterior than with a Gibbs posterior. This approach was used by Dziugaite and Roy [1] for neural network models, and our computational method adapts their approach to LAE models.
> > >
> > > Our Algorithm 1 computes the tightest RH of Eq (14) using S4, not using S3. That is, we search for the optimal $\rho$ within the space of Gaussian distributions. One key advantage of Algorithm 1 is high efficiency, thanks to the closed form solution for $\rho$ provided by Theorem 5.2, which eliminates the need for sampling. As demonstrated in our experiments, Algorithm 1 can efficiently compute a sub-optimal RH on large real-world datasets. This sub-optimal result is sufficient for our purposes, as it successfully validates the non-vacuousness of the bound.
> > >
> > > **On converting the statement from $\forall \rho, P(...)$ to $P(\forall \rho, ...)$**:
> > >
> > > We agree to adopt this conversion for all bounds in Section 2, 3, 4, as we realized that they can be generalized from S2 to S1. We also acknowledge that S1 is a more standard statement in the PAC-Bayes literature [2]. After careful review, we confirm that this change is *independent of* the proofs presented in our paper. Therefore, implementing the change does not affect the correctness or integrity of our theoretical framework, but reveals its more fundamental form.
> > >
> > > However, our bound in Section 5 is not applicable to this conversion, since it is of the S4 form where we explicitly take $\rho$ to be Gaussian.
> > >
> > > **Other issues related to the reviewer's comments**:
> > >
> > > 1. " the KL divergence should be absent in their formulation of the bound... since in all their statements the $\forall \rho$ is outside of the probability". The meaning of "absent" here is ambiguous to us. Additionally, we are unclear why "$\forall \rho$ is outside of the probability" would imply that "KL-Divergence is absent from the bound", as the KL-Divergence is explicitly present in the bound regardless of the position of $\forall \rho$.
> > >
> > > 2. "Adding a positive quantity like the KL divergence in the risk bound still makes it valid but it also makes it much looser". We respectfully disagree with this. Our experimental results show that the KL-Divergence does not make our bound much looser: as shown in the Table 3 of our paper, their values are trivial.
> > >
> > > 3. "I would have never though that they would make this error. But this is what they did!" As mentioned earlier, we agree that replacing $\forall \rho, P(...)$ with $P(\forall \rho, ...)$  is reasonable and will implement this in the final version. We appreciate you pointing out this issue and helping to improve the paper. However, we respectfully disagree referring this issue as an "error", since it does not involve any logical mistake in our proofs. Our theoretical framework remains correct with the revised statement based on your suggestion.
> > >
> > > **References**:
> > >
> > > [1] Dziugaite and Roy. Computing nonvacuous generalization bounds for deep (stochastic) neural networks with many more parameters than training data. arXiv, 2017.
> > >
> > > [2]  Pierre Alquier. User-friendly introduction to pac-bayes bounds. arXiv, 2021.

---

### Official Review · Reviewer_TXNp · 2025-03-10

**Overall Recommendation:** 3

**Summary:**

In this paper, new PAC-Bayesian generalization bounds are derived for multivariate linear regression with Gaussian data or data with bounded support. Specifically, these results are applied to linear autoencoders, which are of practical relevance due to their widespread use in recommender systems. The bound is numerically evaluated for an example with oracle knowledge of the distribution, and compared to the empirical test loss. The bound is found to be reasonably close.

## update after rebuttal

I thank the authors for their response. There has been some issues related to the uniformity of the posterior and the exact nature of the common assumptions of distributions, which would need to be resolved, but these do not seem like critical issues. I retain my current evaluation.

**Claims And Evidence:**

Overall, yes. See discussion of experimental design

**Essential References Not Discussed:**

The paper “PAC-Bayes unleashed: generalisation bounds with unbounded losses” by Haddouche et al includes bounds for linear regression. To some extent, “Information-Theoretic Foundations for Machine Learning” by Jeon and van Roy may be related in that techniques of a somewhat similar flavor are used to study e.g. regression problems.

**Experimental Designs Or Analyses:**

The way in which the oracle parameters are assumed means that the results are not necessarily directly applicable to practical scenarios, but the approach is motivated and discussed in the paper. Also, the discussion of non-vacuousness may not be fully meaningful (see questions below).

**Methods And Evaluation Criteria:**

Yes. The issue of needing oracle knowledge to evaluate the bound makes it somewhat prohibitive, but the way this is addressed in the paper enables some demonstration of the bound.

**Other Comments Or Suggestions:**

There are significant margin overflows in several equations

1. Use \citet when specifically referring to authors
2. Abstract: “Experimental results demonstrates” -> demonstrate
3. In Sec. 5, Eq (25) and Eq (33) are referenced before definition without clarification

**Other Strengths And Weaknesses:**

The paper is overall nicely written and seems to provide a solid theoretical contribution. However, exactly what the bounds are intended to address could be clarified. Since they are oracle bounds, they cannot be used for e.g. estimating test errors in practical settings. In (10), it is argued that one can find the optimal posterior. Can this be compared to normal or regularized linear regression? Also, at present, the conclusions section is simply a summary and all discussion is relegated to the appendix. Moving the key points that should be highlighted to the main paper may be beneficial.

**Questions For Authors:**

1. In several places of the text, it is pointed out that the derived bound is an oracle bound. Is it possible to instead derive an empirical PAC-Bayes bound, as alluded to? Would this be desirable?
2. As mentioned, recommender systems are often evaluated using other metrics than squared loss. Is it possible to at least empirically investigate the correlation of the bound to such metrics?
3. As mentioned in the previous point, non-vacuousness may not be meaningful for the squared loss. Can something like a correlation between the bound and the underlying test loss instead be evaluated to justify the merits of the bound?

**Relation To Broader Scientific Literature:**

The paper extends PAC-Bayesian analysis to a setting where it had previously not been applied. The discussion of prior work on PAC-Bayes is somewhat limited, for instance the fact that the discussion of numerical evaluations is essentially limited to Dziugaite & Roy (2017). However, while there has been follow-up work by the same authors and others, this is perhaps not essential to the present paper.

**Theoretical Claims:**

I did not check the proofs in detail, but found no issues on a skim

---

> ### Author Rebuttal · Authors · 2025-04-01
>
> **Question 1**: Is it possible to instead derive an empirical PAC-Bayes bound, as alluded to? Would this be desirable?
>
> **Answer**: The loss in our linear regression setting is assumed to be unbounded. According to [1], existing empirical PAC-Bayes bounds,  such as Catoni's bound and Seeger's bound, are primarily designed for bounded loss. They cannot be directly applied to our linear regression setting.
>
> Developing an empirical PAC-Bayes bound for unbounded loss is challenging. The reason is that, unbounded losses requires extra assumptions, such as a bounded cumulant generating function (CGF) [2], to characterize their tail behavior. These assumptions typically assume the data distribution is known -- for example, the CGF inherently contains the true risk -- which naturally involves oracle information. As a result, the PAC-Bayes bounds derived under these unbounded loss assumptions are oracle-based.
>
> It remains an important open question that whether unbounded loss assumptions can be made without relying on oracle information, thereby allowing us to derive a fully empirical PAC-Bayes bound.
>
> **Question 2 and Question 3**: As mentioned, recommender systems are often evaluated using other metrics than squared loss. Is it possible to at least empirically investigate the correlation of the bound to such metrics?
>
> **Answer**: Yes. We have added the experimental results of Recall@50 and NDCG@100 using the same test sets and models in Table 2 of our paper. Both Recall@50 and NDCG@100 are recommendation metrics used in the EASE paper [3]. The results are as follows:
>
> Recall@50:
> |$\gamma$ |ML20M|Netflix|MSD|
> |----|----|----|----|
> |$50$| 0.3434 | 0.2567 | 0.3454 |
> |$100$| 0.3453 | 0.2580 | 0.3472 |
> |$200$| 0.3471 | 0.2592 | 0.3486 |
>
> NDCG@100:
> |$\gamma$ |ML20M|Netflix|MSD|
> |----|----|----|----|
> |$50$| 0.4342 | 0.3766 | 0.3187 |
> |$100$| 0.4373 | 0.3785 | 0.3205 |
> |$200$| 0.4402 | 0.3804 | 0.3220 |
>
> Comparing the two tables with Table 2 in our paper, we observe that on each dataset, as $\gamma$ increases, LH and RH decrease, while Recall@50 and NDCG@100 increase. This confirms the expected correlation between LH, RH and Recall@50, NDCG@100, suggesting that our metrics LH and RH correctly reflect the model performance. (Note that LH and RH are losses, smaller value represents better model performance; Recall@50 and NDCG@100 are rankings, larger value represents better model performance.)
>
> Hence, we have empirically validated the correctness of this correlation. We will add these results to our paper. Additionally, we believe that analyzing this correlation theoretically is challenging.
>
> **Essential References Not Discussed**: The paper “PAC-Bayes unleashed: generalisation bounds with unbounded losses” by Haddouche et al includes bounds for linear regression ...
>
> **Answer**: Thanks for sharing the two papers. We will add them into our related works.
>
> In Haddouche et al's paper, we notice that they use the $L_1$ norm to define the loss in linear regression, which is not typical. The $L_1$ norm may help satisfy HYPE, an essential assumption in their PAC-Bayes bounds. However, we believe that the squared $L_2$ (Frobenius) norm we used is a more standard loss for linear regression problems.
>
> **Other Strengths And Weaknesses**: In (10), it is argued that one can find the optimal posterior. Can this be compared to normal or regularized linear regression?
>
> **Answer**: We think that they are not directly comparable. In normal or regularized linear regression, the optimal predictor is a constant, known as a deterministic predictor. In Eq (10), the optimal predictor $W$ is a random variable following the posterior distribution $\rho$, making it a stochastic predictor. Since they represent two different type of predictors, direct comparison between them is not straightforward.
>
> **References**:
>
> [1] Pierre Alquier. User-friendly introduction to pac-bayes bounds. arXiv:2110.11216, 2021.
>
> [2] Borja Rodrıguez-Galvez et al. More PAC-Bayes bounds: From bounded losses, to losses with general tail behaviors, to anytime validity. JMLR, 2024.
>
> [3] Harald Steck. Embarrassingly shallow autoencoders for sparse data. WWW, 2019.

---

### Official Review · Reviewer_zwta · 2025-03-11

**Overall Recommendation:** 3

**Summary:**

This paper provides a PAC-Bayes bound specific to multivariate linear regression. Under Gaussian dataset assumption, the authors derive a sufficient condition for the convergence of said bound. Next, the paper rewrite the PAC-Bayes bound for multivariate linear regression with bounded data distribution and apply it to Linear Autoencoders (LAEs). Finally, the authors derive a method to compute a closed form upper bound of the new PAC-Bayes bound given oracle information (i.e., the prior, data distribution and regularization hyperparameters). This is done in two major steps: finding the optimal posterior distribution and removing the empirical risk in expectation to get a closed form upper bound of the last term in the PAC-Bayes bound.

**Claims And Evidence:**

The claims in the paper are well supported.

**Essential References Not Discussed:**

No.

**Experimental Designs Or Analyses:**

I do not have any issue with the experimental designs and analyses in the experiment section.

**Methods And Evaluation Criteria:**

The calculation of the PAC-Bayes bound involves a grid search of the regularization parameter $\lambda$. I think this is somewhat inefficient since all terms in the upper bound of the PAC-Bayes bound have differentiable closed form formulas (which include the optimal posterior for any given $\lambda$). Therefore, it would be more efficient to use an optimization method such as stochastic gradient descend to find the optimal value for $\lambda$.

**Other Comments Or Suggestions:**

Line 272: Eq(2) $\to$ Eq(3)

**Other Strengths And Weaknesses:**

**Strengths:**
- The paper provides the first PAC-Bayes bound for LAE recommendation systems.
- The paper was able to point out an example where the bound from Shalaeva et al., 2020 diverges and provide a sufficient condition for convergence in the case of Gaussian data.
- The paper have a nice result showing that upper bound when the zero diagonal assumption is applied is bounded by the upper bound when there is no such assumption.
- The paper is fairly well-written and the analyses are rigorous.

**Weaknesses:**
- The data distribution for LAEs in section 4.2 seems restrictive. See question 2 in Questions For Authors.
- Only LAE recommendation systems are considered which is quite limited considering the developed bound applies to any bounded data distribution.
- The PAC-Bayes bound is an oracle bound which is not useful unless the data distribution is known.

**Questions For Authors:**

Question 1: Why is there no convergence analysis for the PAC-Bayes bound in the bounded data case?

Question 2: In section 4.2, the paper seems to assume that each element of the input $x=(x_1,\ldots,x_n)\in \{0,1\}^n$ independently affect its output in $y=(y_1,\ldots,y_n)\in \{0,1\}^n$ i.e., only $x_i$ affect the value of $y_i$ for each $i\in 1,\ldots,n$. Is there a reason for this choice of data distribution?

**Relation To Broader Scientific Literature:**

The PAC-Bayes bound in the paper is a natural extension from the bound for multiple linear regression to multivariate linear regression. It also serves as the first bound for LAE recommendation systems.

**Theoretical Claims:**

I looked at the proofs for the theorems that appear in the main text and find no major issues. I do have an issue with how the paper reuse the $\eta_j$ symbol to mean different things in different sections which is quite confusing.

---

> ### Author Rebuttal · Authors · 2025-04-01
>
> **Question 1**: Why is there no convergence analysis for the PAC-Bayes bound in the bounded data case?
>
> **Answer**: Because the bounded data assumption (Assumption 4.1) alone is not sufficient to guarantee the convergence. The reason is as follows:
>
> First, the convergence of Alquier's bound does not hold for all choices of $\pi$ and $\mathcal{D}$: Our bound in Theorem 1 is a specialization of Alquier's bound, and we have shown in Section 3.2 that certain choices of $(\pi, \mathcal{D})$ can cause divergence.
>
> Next, the bounded data case (Eq (7)) is another specialization of Alquier's bound, derived under Assumption 4.1. Assumption 4.1 imposes a constraint on $\mathcal{D}$, which defines a subset of $(\pi, \mathcal{D})$ in the set of all $(\pi, \mathcal{D})$. However, Assumption 4.1 does not imply that every choice of $(\pi, \mathcal{D})$ in this subset guarantees the convergence.
>
> **Question 2**: In section 4.2, the paper seems to assume that each element $x = (x\_1, x\_2, ..., x\_n) \in \\{0, 1\\}^n$ of the input independently affect its output $y = (y\_1, y\_2, ..., y\_n) \in \\{0, 1\\}^n$ in i.e., only $x_i$ affect the value of $y\_i$ for each $i = 1, 2, ..., n$. Is there a reason for this choice of data distribution?
>
> **Answer**: We would like to clarify that $y = (y\_1, y\_2, ..., y\_n) \in \\{0, 1\\}^n$ is the target, $\hat{y} = (\hat{y}\_1, \hat{y}\_2, ..., \hat{y}\_n) \in \mathbb{R}^n$ is the output of the LAE model $W\in \mathbb{R}^{n\times n}$, with $\hat{y} = Wx$, and $x = (x\_1, x\_2, ..., x\_n) \in \\{0, 1\\}^n$ is the input. The loss between $y$ and $\hat{y}$ is given by $\|\|y - \hat{y}\|\|\_F^2$.
>
> For the output, $\hat{y}\_i$ is not independently affected by $x\_i$ but by the entire vector $x$, since $\hat{y}\_i = W\_{i*}x$.
>
> For the target, we assume that $y_i$ depends only on $x_i$ for simplicity. This assumption is sufficient to describe the constraint that either $x\_i = 1, y\_i = 0$ or $x\_i = 0, y\_i = 1$. While one could assume a more complicated distribution for $y\_i$ (e.g., $y\_i$ depends on the entire $x$) to describe the same constraint, it is unnecessary.
>
> **Theoretical Claims**: I do have an issue with how the paper reuse the $\eta\_j$ symbol to mean different things in different sections which is quite confusing.
>
> **Answer**: We apologize for any confusion caused by the reuse of symbols. The $\eta\_j$ in Section 2 represents an eigenvalue of $\Sigma\_W$, and in Section 5.2, it is redefined as an eigenvalue of $A$. We acknowledge that a clarification regarding this redefinition is missing and will add it to our paper.
>
> **Methods And Evaluation Criteria**: It would be more efficient to use an optimization method such as stochastic gradient descend to find the optimal value for $\lambda$.
>
> **Answer**: It is generally considered impossible for optimizing the right hand side of a PAC-Bayes bound with respect to $\lambda$ [1][2]. The reason is as follows: Let $S$ be the dataset, which is a random variable. In PAC-Bayes bounds, $\lambda$ is assumed to be a constant independent of $S$. However, if optimizing the right hand side of a PAC-Bayes bound with respect to $\lambda$, then the optimal $\lambda$ may depend on $S$, which contradicts the assumption.
>
> There are several ways addressing this issue. One way is the grid search method we used, primarily following [1]. The grid search is simple but may not be efficient. Another way is to derive an upper bound for the RH of Eq (14) that does not depend on $\lambda$, which is used by Rodríguez-Gálvez et al [2].
>
> **References**:
>
> [1] Pierre Alquier. User-friendly introduction to pac-bayes bounds. arXiv:2110.11216, 2021.
>
> [2] Borja Rodrıguez-Galvez et al. More PAC-Bayes bounds: From bounded losses, to losses with general tail behaviors, to anytime validity. JMLR, 2024.

---

> > ### Comment · Reviewer_zwta · 2025-04-04
> >
> > Thank you for your thorough response. My concerns in the first two questions have been addressed. However, I have another question for the last answer as follows:
> >
> > Here is some clarification regarding my comment about optimizing for $\lambda$. While I agree that $\lambda$ cannot be optimized in general, it can be done for the specific model the paper is considering. Since, for any given $\lambda$, the optimal $\rho$ and its related quantities in eq(14) has closed form solution according to Theorem 5.2 and equations (33,15,16,12), the right hand side of the PAC-Bayes bound in eq(14) can be written in terms of $\lambda$. Hence, RH in Algorithm 1 is a function of $\lambda$ and can be minimized using optimization algorithms assuming all other inputs of Algorithm 1 (i.e., $\Sigma_{rr},p,\delta,X,Y,W$) are constant. The concern of Alquier (2021) is, from what I read, that we cannot express the dependency of $\rho$ on $\lambda$. In the case of Algorithm 1, the dependency of $\rho$ on $\lambda$ is explicit and we can simply writing out this dependency. Therefore, we can optimize for $\lambda$ without making $\lambda$ depend on $\rho$ but only in the context of the paper.
> >
> > **Reference:**
> >
> > Alquier, P. User-friendly introduction to pac-bayes bounds. arXiv preprint arXiv:2110.11216, 2021.

---

> > > ### Author Response · Authors · 2025-04-05
> > >
> > > Thanks for your insightful comments! We apologize for not conveying our ideas clearly in our previous answer to the last question.
> > >
> > > We think the main obstacle for optimizing the RH of Eq (14) with respect to $\lambda$ is that the optimal $\lambda$ can depend on the dataset $S$. The reason is as follows: if plugging in Eq (33, 15, 12) and the closed form solution of $\rho$ in Theorem 5.2 to the RH of Eq (14), then the RH of Eq (14) can be expressed as a function $f(\lambda, X, Y , \mathcal{U}\_0, \sigma, \Sigma\_{rr}, \delta, L)$. Denote $S = (X, Y) = \\{(x\_i, y\_i)\\}\_{i=1}^m$ where each $(x\_i, y\_i)$ is i.i.d. from $\mathcal{D}$, so $S$ is a random variable. Consider taking a specific *observation* $(X\_0, Y\_0)$ of $S$ and solving the optimization problem $\lambda^* = \text{argmin}\_{\lambda}\\, f(\lambda | X\_0, Y\_0,  \mathcal{U}\_0, \sigma, \Sigma_{rr}, \delta, L)$ (i.e.,  treating $\lambda$ as the only variable of $f$ by fixing all other variables), then $\lambda^*$ depends on $(X\_0, Y\_0)$. If we plug $\lambda = \lambda^*$ into the RH of Eq (14), then Eq (14) may not hold on another observation $(X\_1, Y\_1)$ of $S$. Such bound is not desirable.
> > >
> > > This issue is mentioned in Example 2.1 of Alquier's paper [1]:
> > >
> > > "Note that the optimization with respect to $\lambda$ is a little more problematic when $\pi$ is not uniform, because the optimal $\lambda$ would depend on the data, which is not allowed."
> > >
> > > And in Section 3.2 of Rodriguez-Galvez's paper [2]:
> > >
> > > "If we could optimize the parameter $\lambda$ in Theorem 11, we would obtain a PAC-Bayes analogue to Chernoff’s inequality for losses with a bounded CGF. However, this is not possible since the optimal parameter depends on the data realization but needs to be selected before the draw of this data."
> > >
> > > We acknowledge that the explanation in Section 2.1.4 of Alquier's paper [1] may be somewhat unclear: It attributes this issue to the optimal $\lambda$ depending on $\rho$. We think the more fundamental reason is that $\rho$ can depend on $S$, as seen in our optimal Gaussian posterior $\rho$ (Theorem 5.2) and the Gibbs posterior $\rho$ in Alquier's paper [1]. Consequently, the optimal $\lambda$ can depend on $S$.
> > >
> > > **References**:
> > >
> > > [1] Pierre Alquier. User-friendly introduction to pac-bayes bounds. arXiv:2110.11216, 2021.
> > >
> > > [2] Borja Rodrıguez-Galvez et al. More PAC-Bayes bounds: From bounded losses, to losses with general tail behaviors, to anytime validity. JMLR, 2024.

---

### Official Review · Reviewer_zmkJ · 2025-03-13

**Overall Recommendation:** 3

**Summary:**

The paper introduces a novel PAC-Bayes generalization bound for multivariate linear regression based on the standard Alquier's (oracle) bound. This result generalizes previous PAC-Bayes bounds for multiple linear regression. After studying conditions that ensure convergence of the bound, they adapt it for Linear Autoencoders and they compute it for several datasets in order to verify its non-vacuousness.

**Claims And Evidence:**

Yes

**Essential References Not Discussed:**

N/A

**Experimental Designs Or Analyses:**

Yes

**Methods And Evaluation Criteria:**

Yes

**Other Comments Or Suggestions:**

1- In the first paragraph of Section 2, along with the standard definitions of empirical risk and so on, you should include the definition of the KL divergence, $D(\rho||\pi)$, which is nowhere to be found in the paper.

2- I've noticed that you optimize the free parameter in the bounds, $\lambda>0$, over a finite grid using union bounds. You should also check the technique in Theorem 12 of [1], which results in potentially tighter optimization of $\lambda$ without too many complications.

[1] Rodríguez-Gálvez, Borja, Ragnar Thobaben, and Mikael Skoglund. "More PAC-Bayes bounds: From bounded losses, to losses with general tail behaviors, to anytime validity." Journal of Machine Learning Research 25.110 (2024): 1-43.

**Other Strengths And Weaknesses:**

I appreciate all the grunt work needed for adapting Alquier's bound to their specific setting. The detailed convergence analysis is also a good touch, since it is often ignored in this kind of papers. The only clear weakness is the experimental part: since the bound is an oracle one, the empirical evaluation needs many compromises and extra assumptions (such as the one in line 422 regarding $\Sigma_{rr}$), which is a bit disappointing. However, I recognize that working out fully empirical PAC-Bayes bounds (especially for unbounded losses) is not easy.

Overall, I believe this is a solid contribution worth publishing.

**Questions For Authors:**

N/A

**Relation To Broader Scientific Literature:**

The paper extends PAC-Bayes bounds for linear regression to more general settings, namely multivariate linear regression and linear autoencoders.

**Theoretical Claims:**

The convergence analysis in Sections 2 and 3.2 seems correct. I haven't carefully checked the proofs in the Appendices.

---

> ### Author Rebuttal · Authors · 2025-04-01
>
> **Question 1**: The KL divergence $D(\rho||\pi)$ is not defined in the paper.
>
> **Answer**: Thanks for pointing out this issue. We will add this into our paper: Suppose $\pi$ and $\rho$ are continuous probability distributions on $\mathbb{R}^d$. The KL-Divergence is defined as
> \begin{equation*}
>     D(\rho||\pi) = \mathbb{E}\_{\theta \sim \rho}\left[\log \frac{\rho(\theta)}{\pi(\theta)}\right]
> \end{equation*}
> where $\rho(\theta)$ and $\pi(\theta)$ denote the probability density function of $\rho$ and $\pi$, respectively. We assume for any $\theta \in \mathbb{R}^d$, $\pi(\theta) = 0$ implies $\rho(\theta) = 0$.
>
> **Question 2**:  You should also check the technique in Theorem 12 of [1], which results in potentially tighter optimization of $\lambda$ without too many complications.
>
> **Answer**: Thanks for sharing Rodríguez-Gálvez et al's work. We find the ideas in this paper both interesting and inspiring.
>
> Based on our understanding, Rodríguez-Gálvez et al address the difficulty of optimizing $\lambda$ by proposing an upper bound that is independent of $\lambda$. Their method constructs this upper bound by partitioning the event space based on the KL-divergence $D(\rho||\pi)$: the event $\mathcal{E} = \\{D(\rho||\pi) \le m\\}$ is split into sub-events $\\{\mathcal{E}\_k\\}_{k=1}^m$ where $\mathcal{E}\_k = \\{D(\rho||\pi) \le k\\}$. Then the bound is optimized by conditioning on each $\mathcal{E}\_k$.
>
> While this is an elegant theoretical contribution, implementing and evaluating it in our context would require significant additional work that is not feasible within the rebuttal period. We plan to reference this approach in our final revision and explore it more thoroughly in future work.
>
> Additionally, we believe our current method based on a grid search of $\lambda$ provides satisfactory results at a low computational cost, especially given our evaluation on large real-world datasets. While a tighter bound may be achievable, it would come at the expense of increased computational cost.
>
> **Other Strengths And Weaknesses**: The only clear weakness is the experimental part: since the bound is an oracle one, the empirical evaluation needs many compromises and extra assumptions (such as the one in line 422 regarding $\Sigma\_{rr}$), which is a bit disappointing. However, I recognize that working out fully empirical PAC-Bayes bounds (especially for unbounded losses) is not easy.
>
> **Answer**: Thanks for your understanding.
>
> Indeed, in our linear regression setting, the loss is unbounded, and we recognize the inherent difficulty in developing fully empirical PAC-Bayes bounds under such conditions. Unbounded loss requires extra assumptions, such as a bounded cumulant generating function (CGF) [1], to characterize their tail behavior. These assumptions typically presuppose the data distribution is known -- for example, the CGF inherently contains the true risk -- which naturally involves oracle information. As a result, the PAC-Bayes bounds derived under these assumptions are oracle-based. Whether unbounded loss assumptions can be made without oracle information remains an important open question, as it is crucial for developing a fully empirical bound.
>
> We would like to highlight an additional strength of our method. In our supplemental experiments addressing the Question 2 from reviewer TXNp, we empirically verified that the decrease in LH and RH of our oracle bound Eq (14) correlates correctly with the increase in Recall@K and NDCG@K. This suggests that LH and RH can correctly reflect model performance. Although LH and RH contain oracle information, they are easier to analyze statically than Recall@K and NDCG@K, making them potentially better evaluation metrics for LAEs.
>
> **References**:
>
> [1] Borja Rodrıguez-Galvez et al. More PAC-Bayes bounds: From bounded losses, to losses with general tail behaviors, to anytime validity. JMLR, 2024.

---

> > ### Comment · Reviewer_zmkJ · 2025-04-04
> >
> > Thank you for you answer. I think there are some nuances to your Answer to **Other Strengths And Weaknesses**. The bounded CGF assumption and related tail assumption (e.g. sub-Gaussianity) don't presuppose that the data distribution is known: they presuppose that **something** about the data distribution is known, namely the behavior of its tails. The data distribution is still unknown, and this is represented by the (unknown) constants bounding the CGF. There are natural scenarios where we know that the loss or other random variable is light-tailed, which allows us to bound the CGF, so your assertion "*These assumptions typically presuppose the data distribution is known*" is not true.

---

> > > ### Author Response · Authors · 2025-04-04
> > >
> > > We appreciate your clarification! We acknowledge that our assertion was indeed incorrect, as the data distribution is not "presupposed to be known". This would be a more accurate statement: *To characterize the tail behavior of an unbounded loss, we typically require a quantity related to the data distribution, such as a CGF. We make assumptions on unbounded loss by assuming such quantity is known, which presupposes that we know some oracle information about the data distribution.*

---

### Decision · Program_Chairs · 2025-05-01

**Decision:**

Reject

**Comment:**

At the end of the discussion period, all reviewers agreed on the "weak acceptance" of the paper. The contribution to the PAC-Bayes literature is theoretically solid, and the assumptions are motivated.

That being said, addressing the multiple clarity issues raised by the reviewers requires a major revision. Notably, Reviewer 8qwC correctly mentions that all PAC-Bayes theorem statements are incorrectly stated (fortunately, it doesn't falsify them) and the experimental procedure lacks clarity. I entirely agree with Reviewer TXNp that the discussion on non-vacuousness is meaningless in the context of the square loss and/or oracle bounds. The already vague term "non-vacuous" was introduced in the PAC-Bayes literature to to celebrate the achievement of zero-one loss bounds lower than 0.5. The discussion on the topic must be amended (I suggest qualifying the bounds as "arguably tight").  More generally, authors' responses alleviate ambiguities and improve the discussion on related work. All of these should make their way to a revised manuscript.

Given the competitive aspect of ICML and its high-quality standard, we decided to reject the manuscript as another review round would be necessary to assess that the final version contains all the unavoidable revisions. I strongly encourage the authors to pursue their great work and submit an improved paper to a future venue.